# A consensus molecular subtypes classification strategy for clinical colorectal cancer tissues

Tim R de Back[1,2,3,]*, Tan Wu[4,5,]*, Pascale JM Schafrat[1,2,3,6], Sanne ten Hoorn[1,2,3], Miaomiao Tan[5,7], Lingli He[5], Sander R van Hooff[1,2,3], Jan Koster[1,2], Lisanne E Nijman[1,2,3], Geraldine R Vink[8,9], Inès J Beumer[10], Clara C Elbers[1,2,3], Kristiaan J Lenos[1,2,3], Dirkje W Sommeijer[1,11,]†, Xin Wang[5,12,13,]†, Louis Vermeulen[1,2,3,]†

**Consensus Molecular Subtype (CMS) classification of colorectal cancer (CRC) tissues is complicated by RNA degradation upon formalin-fixed paraffin-embedded (FFPE) preservation. Here, we present an FFPE-curated CMS classifier. The CMSFFPE classifier was developed using genes with a high transcript integrity in FFPE-derived RNA. We evaluated the classification accuracy in two FFPE-RNA datasets with matched fresh-frozen (FF) RNA data, and an FF-derived RNA set. An FFPE-RNA application cohort of metastatic CRC patients was established, partly treated with anti-EGFR therapy. Key characteristics per CMS were assessed. Cross-referenced with matched benchmark FF CMS calls, the CMSFFPE classifier strongly improved classification accuracy in two FFPE datasets compared with the original CMSClassifier (63.6% versus 40.9% and 83.3% versus 66.7%, respectively). We recovered CMS-specific recurrence-free survival patterns (CMS4 versus CMS2: hazard ratio 1.75, 95% CI 1.24–2.46). Key molecular and clinical associations of the CMSs were confirmed. In particular, we demonstrated the predictive value of CMS2 and CMS3 for anti-EGFR therapy response (CMS2&3: odds ratio 5.48, 95% CI 1.10–27.27). The CMSFFPE classifier is an optimized FFPE-curated research tool for CMS classification of clinical CRC samples.**

## Introduction

Colorectal cancer (CRC) is the third most occurring cancer and the second leading cause of cancer-related mortality worldwide (1). The course of the disease is impacted by vast heterogeneity across patients, as oncogenic mutations occur in numerous combinations (2), and gene-expression profiles vary substantially (3, 4, 5). Tumor heterogeneity not only pertains to cancer cells but also the tumor micro-environment (6), and, hence, response to systemic therapies and patient outcomes (7, 8). Transcriptomic subtyping of CRC covers cancer gene mutations, tumor cell properties, and also the micro-environment, thereby accurately reflecting key elements of cancer heterogeneity (4). To enhance precision medicine, a readily available transcriptomic subtyping pipeline for clinical tumor samples is warranted.

In the past decade, several groups have established gene-expression based classification systems for CRC (7, 9, 10, 11, 12, 13), which were combined into the Consensus Molecular Subtypes (CMSs) (4). Other transcriptomic taxonomies have been developed as well, such as the ColoRectal Cancer Intrinsic Subtypes (14), the intrinsic CMSs (iCMSs) (15), and the Pathway-Derived Subtypes (16). Importantly, all these systems are related and, to date, the CMSs have been most extensively studied (17). In short, CMS1 is immunogenic, microsatellite instable (MSI), and *BRAF* mutated. CMS2 and CMS3 have epithelial features, with WNT and MYC activation in CMS2 and metabolic deregulation in CMS3. CMS4 has mesenchymal properties with TGF-β activation. Notably, the CMSs bear clinical implications. For example, CMS4 shows dismal survival in local CRC and early dissemination. CMS1 has a poor prognosis in metastatic CRC (mCRC) and benefits from immunotherapy, whereas CMS2 has the best prognosis and response to anti-EGFR therapy (17).

Clinical implementation of the CMSs is hampered by several factors. First, gene-expression based subtyping requires bioinformatics expertise, has a long turnaround time and is costly.

---

[1]Cancer Center Amsterdam, Laboratory for Experimental Oncology and Radiobiology, Center for Experimental and Molecular Medicine, Amsterdam, Netherlands [2]Amsterdam Gastroenterology Endocrinology Metabolism, Laboratory for Experimental Oncology and Radiobiology, Center for Experimental and Molecular Medicine, Amsterdam, Netherlands [3]Oncode Institute, Amsterdam, Netherlands [4]Key Laboratory of Genomic and Precision Medicine, Beijing Institute of Genomics, Chinese Academy of Sciences and China National Center for Bioinformation, Beijing, China [5]Department of Surgery, The Chinese University of Hong Kong, Hong Kong SAR, China [6]Amsterdam UMC Location Vrije Universiteit Amsterdam, Department of Medical Oncology, Amsterdam, Netherlands [7]Institute of Translational Medicine, Zhejiang Shuren University, Hangzhou, China [8]Department of Medical Oncology, University Medical Center Utrecht, Utrecht University, Utrecht, Netherlands [9]Department of Research and Development, Netherlands Comprehensive Cancer Organisation, Utrecht, Netherlands [10]GenomeScan B.V., Leiden, Netherlands [11]Flevohospital, Department of Internal Medicine, Almere, Netherlands [12]Li Ka Shing Institute of Health Sciences, The Chinese University of Hong Kong, Hong Kong SAR, China [13]Shenzhen Research Institute, The Chinese University of Hong Kong, Shenzhen, China

Correspondence: l.vermeulen@amsterdamumc.nl
*Tim R de Back and Tan Wu are shared first authorship
†Dirkje W Sommeijer, Xin Wang, and Louis Vermeulen are shared senior authorship

Attempts to circumvent these barriers have been made, such as gene-panel based (NanoString) CMS classification ([18], [19]), immunohistochemical subtyping ([20], [21]), and deep learning classification of stained tumor slides ([22]). Although these approaches reduce either costs, expertise or time, multi-dimensional transcriptome-wide gene-expression data are lacking in each of these methods, limiting additional molecular tumor characterization and, for example, therapy sensitivity analyses. Furthermore, the CMSs were predominantly established on high-quality RNA data derived from fresh-frozen (FF) tumor samples, whereas formalin-fixed paraffin-embedded (FFPE) tissues, yielding highly degraded RNA, are abundant in the clinic and widely available for research. Tumor classification on RNA from FFPE tissues has been shown to be complicated ([23], [24]), and therefore, an FFPE-curated CMS classifier would be valuable for CRC research and future clinical adoption of the CMSs.

In the present study, we aimed to overcome these hurdles by developing a CMS classification pipeline for highly degraded RNA from FFPE tissue, and compared this approach with the random forest (RF) CMSClassifier applied to matched FF-RNA data as a benchmark. We used transcriptome-wide data to be able to also optimize other transcriptomic CRC taxonomies ([14], [15]), integrate novel developments and reuse the data for future research. The newly designed classifier was applied to an independent cohort of mCRC patients, including those treated with anti-EGFR therapy, and key molecular and clinical associations were assessed. Furthermore, we analyzed mutations in 523 cancer-related genes, and correlations of the CMSs with survival and anti-EGFR sensitivity were explored. Our novel CMSFFPE classifier is a research tool and could enhance clinical implementation of CMS subtyping by enabling robust tumor classification of clinically available FFPE samples.

# Results

## Comparing FFPE-RNA isolation and library preparation techniques

To establish an effective classification procedure for FFPE CRC tissue, we optimized RNA isolation and library preparation methods using samples from the discovery and FFPE-RNA application cohorts (Table 1, Fig S1). To this end, we first compared the RNeasy FFPE kit, the ReliaPrep FFPE total RNA miniprep kit, and the AllPrep DNA/RNA FFPE isolation kit. RNA-quality, as determined with the mean percentage of transcripts over 200 bp in length ($DV_{200}$), was worst for the former ($DV_{200}$ = 21.7%), and comparable for the latter two procedures ($DV_{200}$ = 37.8%, $DV_{200}$ = 34.0%, respectively). For further isolations, the AllPrep DNA/RNA FFPE isolation kit was chosen because of efficient co-isolation of DNA and RNA. Next, we determined the most effective library preparation strategy for FFPE RNA-sequencing (FFPE-RNA-seq). Six FFPE CRC samples were sequenced with whole exome sequencing (WES), NEB, FASTSELECT, and TAKARA (see the Materials and Methods section for details). To assess transcript preservation, the transcript integrity number (TIN) scores per sample were calculated per protocol, for the genes from

the RF CMSClassifier (Fig S2A) and the Single-Sample Predictor (SSP) (Fig S2B). The TAKARA protocol yielded the highest TIN-scores, especially in highly degraded samples. Also, the largest proportion of genes from both the RF and SSP classifiers could be recovered with the TAKARA protocol. Hence, QIAGEN's AllPrep DNA/RNA FFPE isolation kit and the TAKARA library preparation protocol were used to sequence additional FFPE samples.

## A CMS classification model for FFPE archival tissue

Next, we compared matched microarray-derived FF and RNA-seq FFPE gene-expression profiles from the low-quality and moderate-quality discovery sets (Tables 1, S1, and S2). We found a high correlation in both the moderate-quality (Fig 1A) and low-quality set (Fig S3A). However, gene-expression levels of RNA-seq FFPE samples were generally lower than of microarray FF samples. As immune, epithelium, and stroma marker genes are all important to assign the CMSs, we evaluated TIN-scores for these gene categories. Notably, we found that immune marker genes were more likely to be degraded than epithelium and stroma marker genes (Figs 1B and S3B). The degradation of immune genes in the low-quality, moderate-quality, and FFPE-RNA application cohort appeared to be cell type specific, rather than a universal feature of immune cells (Fig S4A–C).

Second, we compared the TIN-score of genes in pathways associated with the CMS subtypes in the discovery sets ([4]) and observed differences in RNA-degradation level per pathway in FFPE samples, with lowly and highly degraded pathways relatively equally distributed across CMSs (Figs 1C and S3C). This suggests that the level of transcript degradation is also pathway- and sample-specific, rather than CMS-specific, necessitating the use of well-preserved genes for CMS classification of FFPE samples.

Next, we assessed the degradation level of CMS feature genes from the original CMS classifiers in the moderate-quality discovery set. One-hundred forty-four genes met the criterion of a median TIN-score >20 across samples and were selected for classifier building. The expression of these genes in FF and FFPE samples is strongly correlated (PCC = 0.58, $P$ < 0.001) (Fig 1D). As expected, the TIN-scores for these genes were significantly higher than for all other genes (Fig 1E).

With the selected 144 feature genes a new RF classifier was trained on the training cohort (Table 1), termed the CMSFFPE classifier.

## Validation of the CMSFFPE classification model

Next, we applied the CMSFFPE classifier to the FFPE samples of the discovery cohort (Fig S1) and compared the results with those predicted by the RF and SSP CMSClassifiers, with core CMS calls from network clustering of matched FF microarray samples as benchmark ([4]). The overall accuracy of the CMSFFPE classifier was 63.6% in the low-quality set and 83.3% in the moderate-quality set, while for the original RF CMSClassifier the accuracy was 41.0% and 66.7%, respectively, and for the SSP classifier 54.6% and 66.7%, respectively (Figs 2A and S5A–C). For the CMSFFPE model, the predicted CMS calls from FFPE samples and FF samples in the low-quality and moderate-quality sets were highly concordant ($P$ = 0.026, $P$ = 0.0040, respectively, Binomial Exact test, Fig S5D), with the highest specificity for CMS2 (both

**Table 1. Brief overview of the datasets used in the study.**

| Dataset | Public ID | Sample preservation | n | Stage | FFPE-RNA isolation protocol | FFPE library preparation protocol | Gene-expression platform |
|---|---|---|---|---|---|---|---|
| Discovery cohort – low-quality set | GSE33113 | FFPE and matched FF | 22 | II | RNeasy FFPE kit (QIAGEN) | NEBNext Ultra Directional RNA Library Prep Kit with rRNA depletion (Illumina) | FFPE: HiSeq4000, 150 bp, paired-end (Illumina) |
| | | | | | | | FF: Affymetrix U133 Plus 2.0 chips |
| Discovery cohort – moderate-quality set | GSE33113 | FFPE and matched FF | 12 | II | Allprep DNA/RNA/miRNA Universal kit (QIAGEN) | Truseq RNA Exome Prep kit (Illumina) | FFPE: HiSeq2500, 65 bp, single-end (Illumina) |
| | | | | | RNeasy FFPE kit (QIAGEN) | | FF: Affymetrix U133 Plus 2.0 chips |
| Training cohort | TCGA COAD and READ | FF | 622 | I–IV | NA | NA | Multiple |
| FF-RNA validation cohort | GSE39582 (CIT cohort) | FF | 519 | I–IV | NA | NA | Affymetrix U133 Plus 2.0 chips |
| FFPE-RNA application cohort | GSE267010 | FFPE | 104 | IV | Allprep DNA/RNA FFPE kit (QIAGEN) | SMARTer Stranded Total RNA-Seq Kit v3 Pico (Takara Bio [TAKARA]) | NovaSeq6000, 150 bp, paired-end (Illumina) |

ID, identification; n, number of patients; FFPE, formalin-fixed paraffin-embedded; FF, fresh-frozen; TCGA, The Cancer Genome Atlas; COAD, colon adenocarcinomas; READ, rectum adenocarcinomas; NA, not applicable; CIT, Cartes d'Identité des Tumeurs.

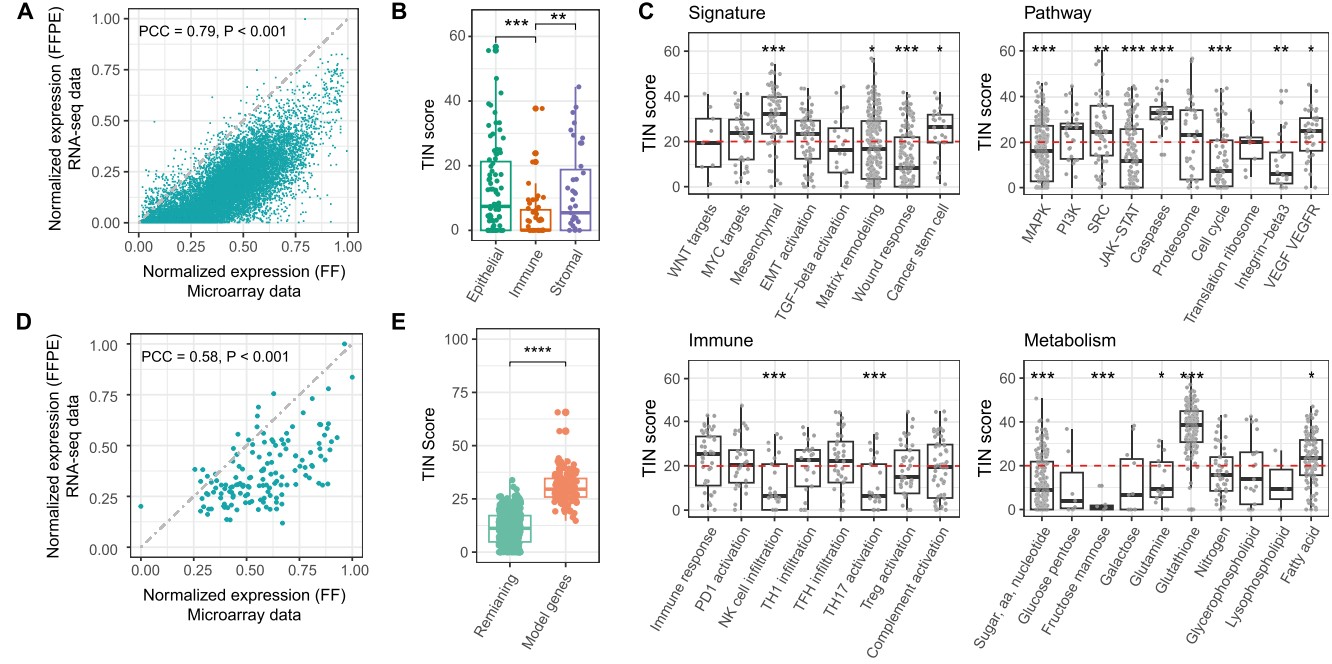

**Figure 1. Overview of the sequencing quality of the moderate-quality discovery set and the selection of feature genes for the CMSFFPE classifier.**
**(A)** The expression values of all expressed genes between matched RNA-seq FFPE and microarray FF samples. **(B)** Box plot showing the TIN-scores of epithelium, immune, and stroma marker genes. **(C)** Box plot showing the TIN-score of genes of different functional pathways. **(D)** The expression distribution of the 144 feature genes for the CMSFFPE classifier between matched RNA-seq FFPE and microarray FF samples. **(E)** TIN-score comparison between the 144 selected feature genes and all remaining genes. *$P < 0.05$; **$P \leq 0.01$; ***$P \leq 0.001$; ****$P \leq 0.0001$.

100%), superior to the RF CMSClassifier and the SSP (Fig S5E and F). Importantly, for the RF CMSClassifier, the pairwise concordance was not statistically significant ($P = 0.58$, $P = 0.073$, respectively, Binomial Exact test, Fig S5E), underlining the need for an FFPE-curated classifier.

To explore the performance of the new CMSFFPE classifier in a larger and high-quality FF-sample dataset, we used microarray-derived RNA data of FF samples from the Cartes d'Identité des Tumeurs (CIT) cohort (FF-RNA validation cohort, Table 1, Fig S1) (11). As a benchmark, we used core CMS calls from network clustering (4).

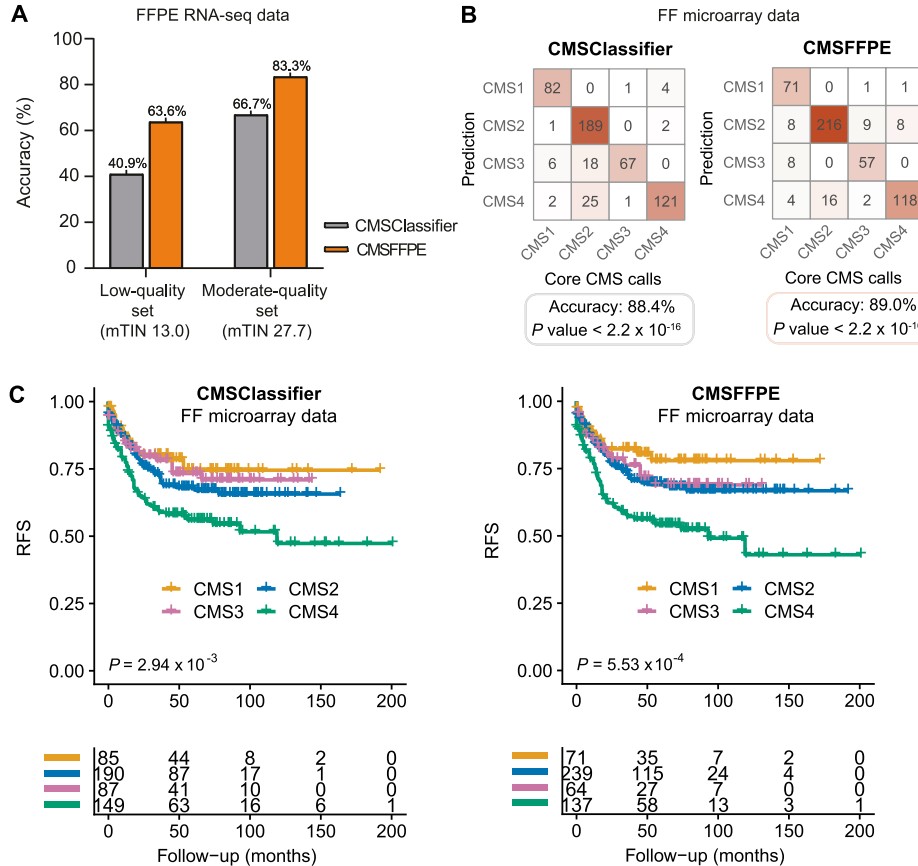

**Figure 2. Validation of the CMSFFPE classifier for FFPE CRC samples.**
**(A)** Bar plot showing the classification accuracy on FFPE-RNA-seq data of the low-quality discovery set and the moderate-quality discovery set for both the CMSClassifier and the CMSFFPE classifier. **(B)** Contingency table showing the concordance of subtypes determined by the CMSClassifier (left) and the RF CMSFFPE classifier (right) in the CIT cohort (microarray FF data). The benchmark CMS calls are core CMS calls from network clustering analyses (4). **(C)** Kaplan–Meier curves of relapse-free survival according to CMS by the RF CMSClassifier (left) and the CMSFFPE classifier (right) in the CIT cohort (microarray FF data).

Subsequently, the CMSFFPE and RF CMSClassifier were applied to predict the CMS subtypes. For both classifiers, the CMS calls were highly concordant with the benchmark calls (Fig 2B). The overall accuracy of the CMSFFPE classifier (89.0%) was similar to the RF CMSClassifier (88.4%) and the SSP classifier (91.1%). Notably, discordant CMS calls between the CMSClassifier and the CMSFFPE classifier manifested as a shift from, respectively, CMS1/3/4 toward CMS2, reflecting the higher accuracy of the CMSFFPE classifier to detect CMS2 (81.5% versus 93.1%). Samples that were discordantly classified showed a significantly lower CMSFFPE probability score, suggesting that these samples had a more CMS-mixed phenotype (Fig S6A–D). Importantly, CMS-dependent differences in relapse-free survival (RFS) could be recapitulated with the new CMSFFPE classifier and were more significant than those identified by the original RF CMSClassifier (Fig 2C). With both the new and the original classifier, CMS1 had the best and CMS4 the worst RFS (Hazard ratio [HR] 2.61, [95% confidence interval (CI) 1.46–4.67]; HR 2.15 [1.29–3.59], respectively).

### Application of the CMSFFPE classifier to the FFPE-RNA application cohort

Because previous studies have provided convincing support for differential anti-EGFR sensitivity across the CMSs (17), we evaluated the performance and clinical relevance of the CMSFFPE classifier in an independent anti-EGFR treated FFPE cohort with a matched control group, consisting of 104 *RAS/BRAF* wt mCRC patients (Table 1, Fig S1). To reflect clinical reality, we obtained either a biopsy or resection specimen of the primary tumor (Table 2). Sixty-two patients were treated with anti-EGFR and these had more distant metastases, which were regularly metachronous, were treated with more lines of systemic treatment and had a worse overall survival (OS) compared with the control group.

As a data-quality check, we assessed the transcriptome mapping and coverage for all samples (Table S3). Most of the samples (90.4%) showed low mapping rates (<50%) (Fig S7A) (25), indicating moderate data quality. Moreover, the fraction of intronic reads was relatively high, ranging from 44.2%–64.5% (Fig S7B). As a consequence of the used library preparation protocol, we found that the read fragments contained large proportions of adapters in the raw sequencing data (Fig S7C), with a median insert size of 72 (Fig S7D). Consistent with the discovery cohort, immune marker genes had the lowest TIN (Fig S7E), and the degradation levels of genes per pathway closely resembled those described for the discovery cohort, despite the different RNA-seq protocols (Fig S7F). In addition, we compared data quality of biopsy versus resections specimens and found a higher quality of biopsy specimens, likely related to a shorter fixation time of biopsy samples (Fig S8A–C). Upon CMSFFPE classification, we identified 4 (3.8%) CMS1 samples, 58 (55.8%) CMS2 samples, 10

**Table 2.   Baseline characteristics of patients included in the FFPE-RNA application cohort.**

| | Total cohort, n = 104 (%) | Treated group, n = 62 (%) | Control group, n = 42 (%) | *P*-value[a] |
|---|---|---|---|---|
| Age (years) median (IQR) | 59 (51.0–68.0) | 61.5 (51.8–68.3) | 57 (50.0–67.0) | 0.189 |
| Sex | | | | |
| Female | 29 (28) | 18 (29) | 11 (26) | 0.751 |
| Male | 75 (72) | 44 (71) | 31 (74) | |
| Performance status | | | | |
| 0 | 50 (48.1) | 26 (41.9) | 24 (57.1) | |
| 1 | 19 (18.3) | 11 (17.7) | 8 (19.0) | |
| 2 | 6 (5.8) | 5 (8.1) | 1 (2.4) | 0.411 |
| 3 | 1 (1.0) | 1 (1.6) | — | |
| Unknown | 28 (26.9) | 19 (30.6) | 9 (21.4) | |
| Specimen | | | | |
| Biopsy | 35 (33.7) | 21 (33.9) | 14 (33.3) | 0.955 |
| Resection | 69 (66.3) | 41 (66.1) | 28 (66.7) | |
| Sidedness | | | | |
| Left | 86 (82.7) | 53 (85.5) | 33 (78.6) | |
| Right | 16 (15.4) | 8 (12.9) | 8 (19.0) | 0.568 |
| Unknown | 2 (1.9) | 1 (1.6) | 1 (2.4) | |
| Metastases | | | | |
| Synchronous | 90 (86.5) | 48 (77.4) | 42 (100) | |
| Metachronous | 13 (12.5) | 13 (21.0) | — | 0.001 |
| Unknown | 1 (1) | 1 (1.6) | | |
| Metastases location | | | | |
| Liver | 92 (88.5) | 57 (91.9) | 35 (83.3) | 0.218 |
| Lung | 38 (36.5) | 32 (51.6) | 6 (14.3) | <0.001 |
| Peritoneum | 26 (25.0) | 16 (25.8) | 10 (23.8) | 0.817 |
| Lymph nodes | 38 (36.5) | 25 (40.3) | 13 (31.0) | 0.330 |
| Bone | 11 (10.6) | 10 (16.1) | 1 (2.4) | 0.047 |
| Other | 18 (17.3) | 14 (22.6) | 4 (9.5) | 0.084 |
| Number of metastases | | | | |
| 0-1 | 47 (45) | 17 (27) | 30 (71) | |
| 2 | 22 (21) | 16 (26) | 6 (14) | |
| 3 | 17 (16) | 14 (23) | 3 (7) | <0.001 |
| >3 | 18 (17) | 15 (24) | 3 (7) | |
| Surgery | | | | |
| Yes | 68 (65) | 40 (65) | 28 (67) | 0.821 |
| No | 36 (35) | 22 (35) | 14 (33) | |
| Chemotherapy | | | | |
| Yes | 102 (98.1) | 62 (100) | 40 (95.2) | 0.161 |
| No | 2 (1.9) | — | 2 (4.8) | |
| Number of treatment lines | | | | |
| None | 2 (1.9) | — | 2 (4.8) | |
| 1 | 42 (40.4) | 8 (12.9) | 34 (81.0) | |
| 2 | 29 (27.9) | 24 (38.7) | 5 (11.9) | <0.001 |
| 3 | 15 (14.4) | 15 (24.2) | — | |
| >3 | 16 (15.4) | 15 (24.2) | 1 (2.4) | |
| Follow-up (months) median (95% CI) | 43.0 (38.9–47.1) | 39.1 (30.7–47.6) | 44.5 (NA) | 0.070 |

**Table 2.   Continued**

| | Total cohort, n = 104 (%) | Treated group, n = 62 (%) | Control group, n = 42 (%) | *P*-value[a] |
|---|---|---|---|---|
| Vital status | | | | |
| Alive | 36 (34.6) | 14 (22.6) | 22 (52.4) | 0.002 |
| Deceased | 68 (65.4) | 48 (77.4) | 20 (47.6) | |

n, number of patients; IQR, interquartile range; 95% CI, 95% confidence interval; NA, not available.
[a]*P*-values are calculated with the Chi-square test and Fisher's Exact test for categorical variables, the Mann-Whitney *U* test for median age comparison between groups and the log-rank test for median follow-up comparison between groups.

(9.6%) CMS3 samples, and 32 (30.8%) CMS4 samples conform the expected distribution in *RAS*/*BRAF* wt mCRC patients (26).

### Molecular characteristics of the CMSs in the FFPE-RNA application cohort

To confirm that the CMSs in the FFPE-RNA application cohort were molecularly similar to the original CMSs, we performed molecular characterization per CMS (Fig 3). First, differential expression analyses of classifier genes showed proper segregation of the CMSs, confirming robust performance of the CMSFFPE classifier (Fig 3A). Next, we assessed gene set enrichment across the CMSs and could recover all previously reported CMS features (Fig 3B). These included enrichment for immune activation in CMS1, SRC signaling, cell cycle and DNA repair in CMS2, metabolic processes in CMS3 and mesenchymal differentiation, EMT, and TGF-*β* signaling in CMS4 (27). Furthermore, the immune cell signature was enriched in CMS1, differentiated enterocytes in CMS2, goblet cells in CMS3, and mesenchymal cells in CMS4.

In addition, we performed mutational characterization with the TruSight Oncology 500 (TSO500) assay and found that patients were most frequently *TP53*- (75%) and *APC*-mutated (57%), probably related to the *RAS*/*BRAF* wt status and predominant left-sidedness of the cohort (Fig 3C). Two patients with MSI tumors were classified as CMS1 and CMS4. Compared with the other CMSs, CMS2 tumors were enriched for canonical mutations in *TP53* (89.1%, *P* = 0.002), and were relatively more frequently mutated in *APC* (63.6%, *P* = 0.312), *TP53* and *APC* combined (58.2%; *P* = 0.321), and *EGFR* (21.8%; *P* = 0.085) (Table S4). Altogether, these findings corroborate robust CMS classification with the CMSFFPE classifier.

### Clinical characteristics per CMS in the FFPE-RNA application cohort

Furthermore, we assessed whether clinical associations of the CMSs could be recovered in our FFPE-RNA application cohort. In line with the *RAS*/*BRAF* wt status of the cohort, most tumors were left-sided, except for CMS1 (75% right-sided) (Table 3). CMS2 patients nearly all had liver metastases (CMS2 versus rest: *P* = 0.004; Chi-square test) and a relatively long OS (median OS 42.9 mo [38.5–47.4]), whereas CMS3 tumors showed the worst OS (median OS 20.7 mo [10.7–30.6]) (data not shown). This might be related to the rather large proportion of advanced CMS3 patients with synchronous metastases compared with the other subtypes. Strikingly, CMS4 tumors performed best with an OS comparable with CMS2 cancers (median OS 47.6 mo [37.9–57.3]). We also assessed OS for the anti-EGFR treated

and untreated subgroups separately and found a similar pattern in both groups, with CMS2 and CMS4 having a comparable OS (HR 0.86 [0.45–1.66], HR 1.07 [0.39–2.90], respectively) (Fig S9A and B). In univariable and multivariable Cox regression models, the CMSs did not independently predict OS (*P* = 0.183), although OS patterns resembled known prognostic associations of the subtypes in mCRC (17), with CMS2 having a better OS than the other subtypes (Table S5, Fig S10).

### Molecular predictors of anti-EGFR response

As a proof of principle for the feasibility of FFPE-RNA and -DNA based precision medicine, we compared mutations between patients that responded to anti-EGFR therapy and patients that did not respond (Fig 4A). We found a correlation between anti-EGFR response and *TP53* mutations (96.2% in responders versus 63.6% in non-responders, *P* = 0.017) (Table S6). Strikingly, the combination of a *TP53* and *APC* mutation was even stronger associated with anti-EGFR response (*P* = 0.004).

Because previous literature suggested a benefit of anti-EGFR therapy for CMS2 tumors, we assessed whether the same patterns could be observed in our FFPE-RNA application cohort. Because CMS1 and CMS3 tumors were scarce in the treated group, we compared anti-EGFR response rates of epithelial (CMS2&3) versus immune and mesenchymal (CMS1&4) tumors, and observed a (non-significant) pattern of a better objective response rate (ORR) for epithelial tumors (ORR 62.5% versus 38.9%, respectively) (Fig 4B). Interestingly, two right-sided tumors responded to anti-EGFR therapy, which were both epithelial tumors with co-occurring *APC* and *TP53* mutations. Among non-responders were six right-sided tumors, of which two were epithelial and two had co-occurring *APC* and *TP53* mutations. After univariable and multivariable logistic regression analyses, independent predictors for anti-EGFR therapy responsiveness (ORR) were CMS2&3 tumors (*P* = 0.038) and co-occurring *TP53* and *APC* mutations (*P* = 0.007) (Table S7, Fig 4C). This suggests that FFPE-based CMS classification and gene-panel DNA sequencing are robust methods to detect distinct patient groups sensitive to systemic therapies.

## Discussion

In this study, we aimed to facilitate the use of clinical CRC samples in transcriptome-wide research into the CMSs by improving the classification strategy for clinically abundant FFPE samples, which could ultimately enhance clinical implementation. We selected 144

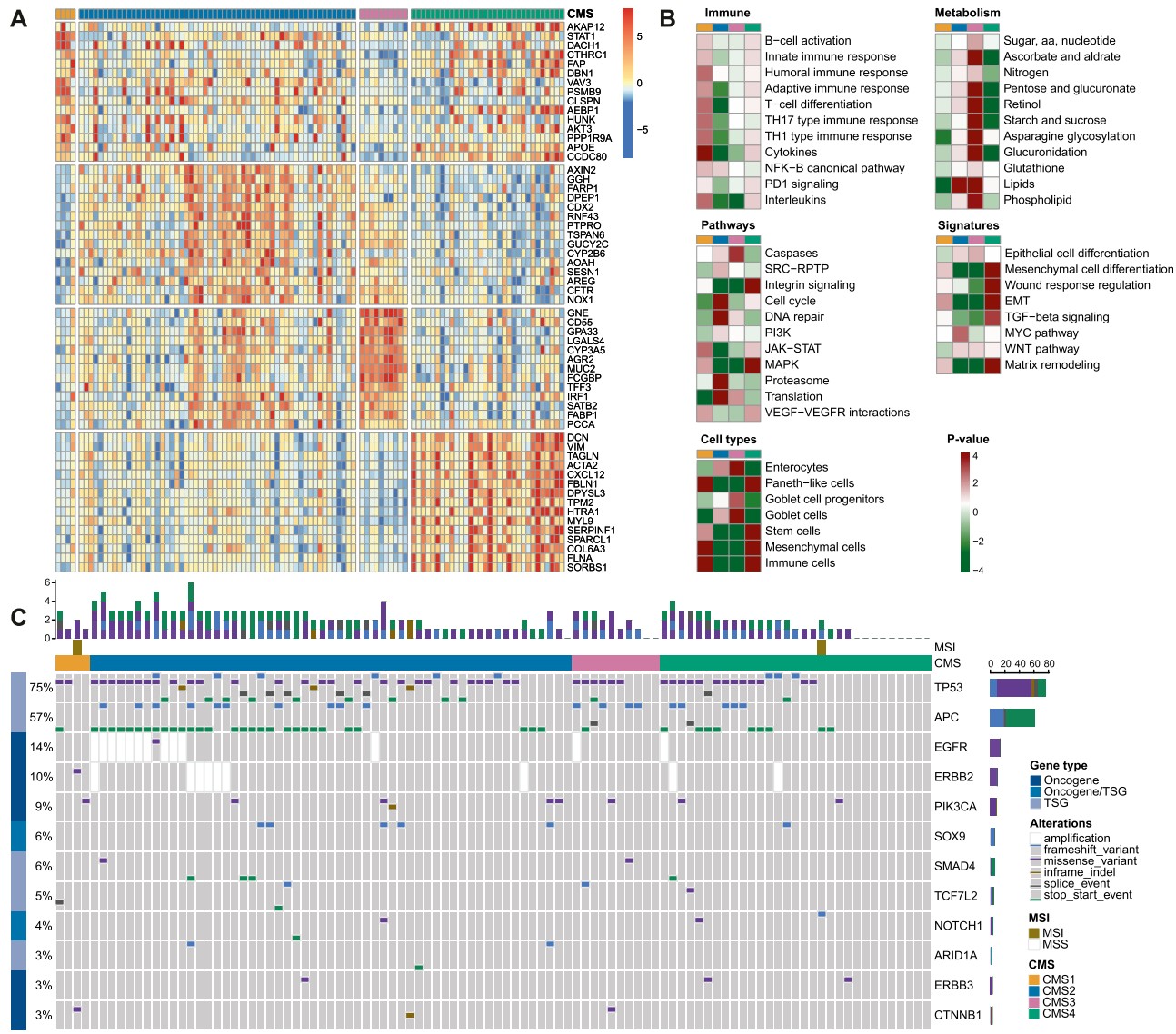

**Figure 3. Molecular characteristics of the CMSs in the FFPE application cohort.**
**(A)** Heatmap showing the 15 top differentially expressed genes between each CMS and the remaining subtypes (RNA-seq data). **(B)** Gene set enrichment analyses for each CMS versus the remaining subtypes for immune sets, metabolism sets, pathway sets, cancer signature sets, and cell type sets (RNA-seq data). **(C)** Oncoplot showing the top ten mutated genes in the FFPE application cohort, grouped by CMS.

well-preserved CMS feature genes in FFPE-derived RNA to train a new CMS classifier. As compared with the original CMSClassifier, the CMSFFPE classifier strongly improved classification accuracy in two FFPE cohorts, and retained high accuracy when classifying FF samples (11). Importantly, known associations of the CMSs with relapse-free survival (RFS) were confirmed in this cohort. Furthermore, in an independent FFPE-RNA application cohort, we could recover key molecular and clinical associations of the CMSs, including enrichment for immune activation and MSI in CMS1, enterocytes, cell cycle, DNA repair, and canonical mutations in CMS2, metabolic deregulation and goblet cell enrichment in CMS3, EMT, TGF-β signaling, and mesenchymal cells in CMS4, predilection of the CMSs for sidedness and metastatic site, and anti-EGFR sensitivity. As a proof of principle, we found that CMS2&3 tumors

were independently associated with response to anti-EGFR therapy.

Degraded RNA, stemming from FFPE samples, significantly impairs accurate gene-expression profiling, and hence, subtyping (23, 24). Although measures like the RNA Integrity Number and the $DV_{200}$ have been proposed to account for low-quality RNA, these metrics require pre-sequencing sample selection and are sample-specific, instead of transcript-specific (28). Inevitably, these methods lead to exclusion of low-quality samples, making them less suitable for clinical application. Here, we used the transcript-specific TIN-score and found that the integrity of RNA varies per functional pathway (e.g., immune-related genes versus epithelial and stromal genes). Furthermore, the variation in stability of RNA transcripts per biological pathway was similar between unrelated cohorts. Pathway-dependent

**Table 3. Characteristics per CMS in the FFPE-RNA application cohort classified with the CMSFFPE classifier.**

| | CMS1, n = 4 (%) | CMS2, n = 58 (%) | CMS3, n = 10 (%) | CMS4, n = 32 (%) | $P$-value[a] |
|---|---|---|---|---|---|
| Age (years) median (IQR) | 48.5 (45.0–68.5) | 59.0 (52.0–68.0) | 58.0 (52.3–63.0) | 61.0 (50.0–69.0) | 0.573 |
| Sex | | | | | |
| Female | 1 (25.0) | 18 (31.0) | 3 (30.0) | 7 (21.9) | 0.877 |
| Male | 3 (75.0) | 40 (69.0) | 7 (70.0) | 25 (78.1) | |
| WHO performance status | | | | | |
| 0 | 3 (75.0) | 25 (43.1) | 8 (80.0) | 14 (43.8) | |
| 1 | 1 (25.0) | 9 (15.5) | 1 (10.0) | 8 (25.0) | |
| 2 | — | 4 (6.9) | — | 2 (6.3) | 0.669 |
| 3 | — | 1 (1.7) | — | — | |
| 4 | — | — | — | — | |
| Unknown | — | 19 (32.8) | 1 (10.0) | 8 (25.0) | |
| Sidedness | | | | | |
| Left | 1 (25.0) | 48 (82.8) | 9 (90.0) | 28 (87.5) | |
| Right | 3 (75.0) | 8 (13.8) | 1 (10.0) | 4 (12.5) | 0.087 |
| Unknown | — | 2 (3.4) | - | — | |
| MMR status | | | | | 0.041 |
| MSI | 1 (25.0) | — | — | 1 (3.1) | |
| MSS | 3 (75.0) | 58 (100) | 10 (100) | 31 (96.9) | |
| Metastases | | | | | 0.590 |
| Synchronous | 1 (25.0) | 51 (87.9) | 9 (90.0) | 27 (84.4) | |
| Metachronous | 3 (75.0) | 6 (10.3) | 1 (10.0) | 5 (15.6) | |
| Unknown | — | 1 (1.7) | — | — | |
| Location of metastases | | | | | |
| Liver | 3 (75.0) | 56 (96.6) | 7 (70) | 26 (81.3) | 0.010 |
| Lung | 2 (50.0) | 25 (43.1) | 2 (20.0) | 9 (28.1) | 0.323 |
| Peritoneum | 2 (50.0) | 12 (20.7) | 1 (10.0) | 11 (34.4) | 0.194 |
| Lymph nodes | 3 (75.0) | 23 (39.7) | 3 (30.0) | 9 (28.1) | 0.274 |
| Bones | — | 7 (12.1) | 2 (20.0) | 2 (6.3) | 0.546 |
| Other | — | 11 (19.0) | 3 (30.0) | 4 (12.5) | 0.529 |
| Surgery of primary | | | | | |
| Yes | 4 (100) | 36 (62.1) | 3 (30.0) | 25 (78.1) | 0.017 |
| No | — | 22 (37.9) | 7 (70.0) | 7 (21.9) | |
| Chemotherapy | | | | | |
| Yes | 4 (100) | 36 (62.1) | 3 (30.0) | 25 (78.1) | 0.017 |
| No | — | 22 (37.9) | 7 (70.0) | 7 (21.9) | |
| Number of treatment lines | | | | | |
| None | — | — | — | 2 (6.3) | |
| 1 | 3 (75.0) | 22 (37.9) | 4 (40.0) | 13 (40.6) | |
| 2 | — | 18 (31.0) | 4 (40.0) | 7 (21.9) | 0.419 |
| 3 | — | 8 (13.8) | — | 7 (21.9) | |
| >3 | 1 (25.0) | 10 (17.2) | 2 (20.0) | 3 (9.4) | |
| Anti-EGFR treated | | | | | |
| Yes | 1 (25.0) | 35 (60.3) | 7 (70.0) | 19 (59.4) | 0.561 |
| No | 3 (75.0) | 23 (39.7) | 3 (30.0) | 13 (40.6) | |

n, number of patients; IQR, interquartile range; WHO, World Health Organization; MMR, mismatch repair; MSI, microsatellite unstable; MSS, microsatellite stable; EGFR, endothelial growth factor receptor.

[a]$P$-values are calculated with the Fisher-Freeman Halton Exact test for categorical variables and the Kruskal-Wallis test for median age comparison between groups.

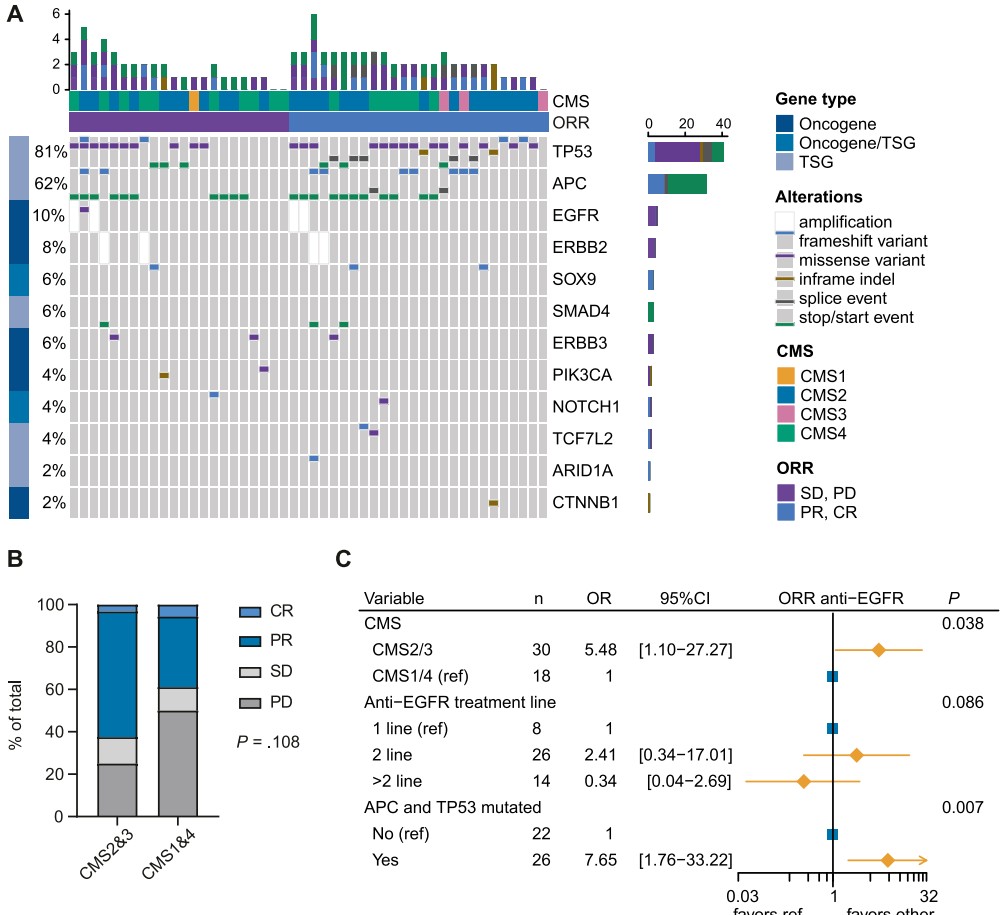

**Figure 4. Anti-EGFR response in the FFPE application cohort.**
**(A)** Oncoplot showing the top ten mutated genes in the anti-EGFR treated group of the FFPE application cohort, grouped by non-response and response. **(B)** Barplot showing the anti-EGFR response for CMS2&3 versus CMS1&4.
**(C)** Multivariable logistic regression analysis for overall response rate to anti-EGFR in patients of the FFPE application cohort.

differences in RNA-degradation levels have been described before and might be related to transcript size and AU- versus GC-nucleotide content (28, 29, 30). Altogether, this necessitated the selection of well-preserved genes in FFPE samples for the CMSFFPE classifier, which significantly increased the accuracy of CMS classification in our two FFPE datasets compared with the original CMSClassifier.

Although the new CMSFFPE classifier significantly improved CMS classification of low-quality RNA samples, a gap in classification accuracy could still be observed between the low-quality (accuracy 63.6%) and moderate-quality sets (accuracy 83.3%). Differences in library preparation strategies between these two datasets might explain disparities in data quality and, consequently, classification accuracy. Whereas samples of both datasets were ribosomal RNA (rRNA) depleted, enrichment for coding RNAs was only performed on samples of the moderate-quality set (exome capture). A previous comparative analysis showed that exome-capture protocols are superior to other methods for sequencing of degraded RNA (31), which is in line with our findings. These results indicate that CMSFFPE subtyping on degraded RNA is more accurate after library preparation with rRNA depletion and exome capture.

CMSs classified by the CMSFFPE classifier showed strong associations of the subtypes with RFS in the CIT cohort, and displayed known characteristics in the FFPE-RNA application cohort (4), which

further confirmed its reliability and clinical utility. We identified an enrichment for liver metastases in CMS2, conform previous reports (32). Two MSI patients were classified as CMS1 and CMS4, as would be expected (4). Right-sided tumors were most common in CMS1 patients (33). However, prognostic associations of the CMSs for OS were not significant in our FFPE-RNA application cohort. This might be attributable to the limited sample sizes per CMS, precluding the detection of significant associations. Second, this cohort was selected for RAS/BRAF wt mCRC patients, and thus not a true reflection of the general CRC population. Interestingly, the observed OS patterns per CMS closely resembled OS per CMS of patients included in the PanaMa trial, in which RAS wt mCRC patients treated with or without panitumumab were studied (34). Hence, the observed OS patterns appear as intrinsic features of the selected patient group and further strengthen the robustness of the CMSFFPE classifier.

We found a high proportion of somatic TP53 and APC mutations in our FFPE-RNA application cohort. This is most likely because of the RAS/BRAF wt status of the patients, thereby enriching for canonical TP53 and APC mutations, as was also observed in other studies (26, 34). In addition, enrichment for TP53 and APC mutations was also found in iCMS2 tumors, which are mostly left-sided and microsatellite stable (MSS), similar to tumors in our cohort, which could support our findings (15).

We observed ORRs to anti-EGFR therapy per subtype comparable with the ORRs found in the PanaMa trial (CMS2 68.9%, CMS3 50%) (34). ORRs for anti-EGFR therapy were lower in the PICCOLO trial (CMS2/3 40.8%) and higher in the COIN trial compared with our study (CMS2/3 77.8%) (35), most probably because of differences in setting (second-line versus first-line versus first-to-fourth-line setting, resp.). In addition, sensitivity of CMS2, and to a lesser extent CMS3, to anti-EGFR therapy has been shown before (17, 35), and underlines the clinical relevance of our classifier. We also found a significant association between co-occurring *TP53* and *APC* mutations and anti-EGFR response. These results correspond with previous studies showing a favorable response to anti-EGFR in CRCs with co-occurring *TP53* and *APC* mutations (36, 37).

The added value of the CMSFFPE classifier lies in the fact that it uses transcriptome-wide data to classify patients. Although alternative FFPE CMS classification strategies have been developed before, such as immunohistochemistry-based and image-based CMS detection (20, 22), these methods lack the acquisition of transcriptome-wide data. These data allow for additional molecular tumor characterization on demand, analyses into therapy sensitivity, and reuse of data for future research. Furthermore, it facilitates incorporation of novel developments and identification of other CRC subtypes beyond the CMSs, as using different CRC taxonomies in parallel is probably most informative for personalized medicine (16). With the advent of quick and more affordable next-generation sequencing techniques (38, 39), and ready-to-use bioinformatics platforms for discovery and classification of complex transcriptomic data (40, 41, 42), clinical tumor classification based on whole transcriptome data comes within reach. Our FFPE classification pipeline could be exploited in future research to facilitate this process and further disentangle clinical decision-making based on tumor molecular properties.

Our study was affected by different gene-expression profiling methods between patient-matched FF and FFPE samples of the discovery cohort (microarray versus RNA-seq). This might have biased the comparison of the CMSFFPE classifier with the original RF CMSClassifier, although this effect is probably limited as studies have demonstrated high concordance between gene-expression data and molecular subtypes originating from microarray analysis versus RNA-seq (24, 43). In addition, patient-matched FF and FFPE samples were not spatially matched within a tumor, which may have underestimated the accuracy of our CMSFFPE classifier, as considerable intra-tumor CMS heterogeneity depending on sampling region has been previously demonstrated (44, 45). Despite differences in transcriptomic profiling method and sampling region between patient-matched FF and FFPE samples, the FF-FFPE concordance in CMS calls of the CMSFFPE classifier was high in the discovery cohorts. Moreover, we also demonstrated similar accuracy of the CMSFFPE classifier compared with the original CMSClassifier in the high-quality FF-RNA CIT cohort, where methodological and sampling region differences do not play a role. Intra-tumor CMS heterogeneity might have also affected CMS calls of biopsy specimens in the FFPE-RNA application cohort, as these cover only small areas of the tumor and are therefore stronger affected by sampling region. Despite including 35 biopsies in this cohort, key CMS characteristics were convincingly recovered, suggesting limited bias through the inclusion of biopsy samples.

Furthermore, the treatment group of the FFPE-RNA application cohort was relatively small, which precluded demonstration of known predictive associations for anti-EGFR therapy per CMS and led to grouping of CMS2 and CMS3 samples (epithelial tumors) versus CMS1 and CMS4 samples (immune and mesenchymal tumors). We did observe a tendency of a superior response to anti-EGFR of CMS2 patients exclusively, but this was not statistically significant. Lastly, while the CMSFFPE classifier was highly accurate in classifying CRC samples of an FFPE cohort, an SSP classifier should be developed to reliably classify single samples.

Here, we developed a new CMS classification tool, the CMSFFPE classifier, specifically designed for accurate classification of transcriptome-wide data derived from FFPE samples. The used method could be used to also optimize other transcriptome-wide classification systems, allows for post-hoc integration of novel developments and reuse of transcriptomic data for future research. We demonstrated a high classification accuracy of the CMSFFPE classifier in two FFPE-RNA datasets and retained classification accuracy in a high-quality FF-RNA dataset, as compared with the original CMSClassifier. We confirmed key molecular and clinical associations of the CMSs in an independent FFPE-RNA application cohort classified with the CMSFFPE classifier, illustrating its robust performance. Additional validation of the classifier in an FF and FFPE patient-matched RNA-seq cohort is warranted. As proof of principle for transcriptome-based precision medicine using FFPE samples, we demonstrated predictive value for anti-EGFR therapy of CMS2 and CMS3 tumors. The new CMSFFPE classifier facilitates CMS classification of clinical FFPE tumor samples in a research context. The classifier could ultimately enhance the process toward clinical implementation of personalized medicine for CRC patients.

# Materials and Methods

### Patient series

Five patient cohorts were used for, respectively, selection of well-preserved genes in FFPE samples (discovery cohort – moderate-quality set), classifier training (training cohort), classifier testing (discovery cohort – low- and moderate-quality sets), classifier application on FF-derived RNA data (FF-RNA application cohort), and classifier application on FFPE-derived RNA data (FFPE-RNA application cohort) (Table 1). The datasets were used in multiple stages of the study (Fig S1).

The discovery sets consisted of patients with localized CRC that underwent curative surgery (6) (Table 1). Clinical information, including survival data, was available. For all discovery patients, both FFPE-derived and FF-derived RNA were available, originating from RNA-seq and microarray analysis, respectively. FF-derived core CMS calls from network clustering were used as benchmark (4). Well-preserved genes in FFPE samples were selected, which were used to construct the CMSFFPE classifier.

The cohorts of colon adenocarcinomas (COAD) and rectal adenocarcinomas (READ) from the Cancer Genome Atlas (TCGA) were used for classifier training (training cohort) (2) (Table 1). CMS calls from the original RF CMSClassifier served as benchmark.

As a microarray-derived FF-RNA validation cohort, the CIT cohort was used (Table 1). This cohort consisted of 519 CRC patients that underwent surgery of the primary tumor, with documented RFS and core CMS calls from network clustering (4). The core CMS calls were used as benchmark for comparing the accuracy of the CMSFFPE classifier with the original RF CMSClassifier.

As an FFPE-RNA application cohort, a total of 104 FFPE tumor biopsy or resection samples from unique *RAS/BRAF* wt mCRC patients diagnosed from January 2015 through December 2019 were requested from the Prospective Dutch CRC cohort study (PLCRC) (46). The sample size gave >95% certainty to detect at least one molecular variant with a prevalence of 3% in the total cohort (power calculation: $0.97^n \leq 0.05$; n = sample size). All patients provided written informed consent for usage of their clinical data and tumor specimens. PLCRC was registered at ClinicalTrials.gov (NCT02070146) and approved by the Medical Research Ethics Committee of the University Medical Centre Utrecht (NL47888.041.14). PLCRC is conducted in accordance with the declaration of Helsinki (46).

Patients were selected based on previous treatment with anti-EGFR (the treatment population), and no previous treatment with anti-EGFR (the control population), and matched on age, sex, ASA-classification, stage, and sidedness. Anonymized clinical data were obtained via the Netherlands Cancer Registry (47). Clinical data consisted of patient characteristics (age, sex, survival), tumor characteristics (stage at diagnosis, metastases, lymph node involvement), and treatment characteristics (surgery, chemotherapy, targeted therapy, documented best response to therapy). Clinical data and survival were last updated in March 2022.

### FFPE-RNA isolation

We compared three different FFPE-RNA isolation procedures on RNA quality, as estimated with the $DV_{200}$, determined by the 4200 TapeStation System (Agilent). We compared the RNeasy FFPE kit (QIAGEN), the ReliaPrep FFPE total RNA miniprep kit (Promega) and the AllPrep DNA/RNA FFPE isolation kit (QIAGEN). For the FFPE-RNA application cohort, total RNA and genomic DNA were isolated from FFPE samples with QIAGEN's QIAcube classic (cat. no. 9001293; QIAGEN). A $DV_{200}$>15% was deemed sufficient for RNA-seq.

### The transcript integrity number

We assessed the transcript preservation levels of our FFPE datasets by calculating the TIN (28). The TIN reflects the percentage of a transcript with uniform read coverage. The TIN-score was calculated for each sample using RSeQC's tin.py (v.5.0.1) for the annotated transcripts (48).

### RNA-sequencing and targeted exome sequencing

To account for variations in FFPE-RNA-sequencing methods, two different procedures were used for FFPE samples of the discovery cohort, dividing the cohort into a low-quality and moderate-quality set, as objectified with the median TIN-score (Tables 1, S1, and S2).

Before sequencing the FFPE-RNA application cohort, four library preparation strategies were compared in six CRC FFPE samples, to determine the impact of library preparation strategy on data quality. We compared (A) whole exome sequencing, (B) NEB Ultra II Directional library preparation with rRNA depletion (NEB), (C) NEB cDNA module with rRNA depletion, Agilent SureSelect All Exon V7 library preparation (FASTSELECT) and (D) SMARTer Stranded Total RNA-seq Kit v3 – Pico from TakaraBio (TAKARA). The library preparation technique yielding the optimal RNA-data quality was applied to the full cohort.

The TSO500 High-Throughput assay was performed on genomic DNA of the FFPE-RNA application cohort as instructed by Illumina. Paired-end next-generation sequencing was performed on the NovaSeq6000 by GenomeScan. The TSO500 detects somatic variants across 523 cancer-relevant genes, as well as tumor mutational burden (TMB) and mismatch repair (MMR) status. Classification of TMB-high or TMB-low was based on a cut-off value of 10 mutations per megabase. MMR status was determined as a percentage based on the analyses of 130 MSI marker sites. A cut-off of 15% was chosen to distinguish MSI-high from MSI-low.

### RNA-data processing and gene-expression quantification

For the discovery cohort, fragments were mapped to the human genome (GRCh38) using STAR v2.7.8a with GENCODE v33 and default parameters (49). Only uniquely mapped fragments were considered and transcripts per million (TPM) were calculated using RSEM v1.3.3 (50). This methodology was similar to TPM quantification performed on TCGA data, which was used for classifier training. Hence, we excluded bias because of differences in TPM quantification methods in the datasets used for feature selection and classifier training.

For the FFPE-RNA application cohort, the RNA-seq data were analyzed using the Cogent next-generation sequencing Analysis Pipeline (CogentAP 2.0), as suggested by TakaraBio (https://www.takarabio.com/products/next-generation-sequencing/bioinformatics-tools/cogent-ngs-analysis-pipeline). In brief, we removed sequencing adapters and trimmed the RNA-seq reads using cutadapt (51) (v3.4, settings: -m 15 --trim-n --max-n 0.7 -q 20). Next, the retained RNA-seq reads were mapped to the human reference genome (GRCh38) and the reads were annotated with GENCODE v33 using STAR (49). The number of uniquely mapped reads per gene per sample was calculated using featureCounts (v2.0.0) (52). Of note, RSEM and featureCounts to quantify TPM were previously shown to be highly concordant (53). Gene count matrices were imported into R (v4.0.3) and converted into TPM with the genomeutils package (https://github.com/ssarda/genomeutils). The Picard software (https://broadinstitute.github.io/picard/) (v2.27.3) was used to estimate the insert sizes of reads.

### Redesigning the CMS classifier

The CMSFFPE classifier was developed with the RandomForest R package (54). The gene selection and model building steps are as follows:

(i) Gene selection: signature genes of the CMSClassifier (4) and CMScaller (55) with a median TIN-score >20 (56) in the moderate-quality set were kept as features for the CMSFFPE classifier.

(ii) Training set: the classifier was trained on the TCGA-CRC cohort with core CMS calls from network clustering (4) as benchmark. TPM values were downloaded with the R package "TCGAbio-links" (57) in "harmonized" mode.

(iii) Model building: to account for variations in gene expression caused by distinct experimental protocols, we first performed z-normalization on the gene-expression profiles, and then derived binary gene pairs from the standardized expression values. This methodology has been previously used for ovarian cancer subtyping (58).

(iv) Validation in FFPE and FF-RNA datasets: to evaluate the performance of the CMSFFPE classifier, we assessed the concordance between the CMS calls predicted from FFPE samples and paired FF samples in the discovery cohort, and compared them with the calls derived from the original CMSClassifier. Next, we applied the CMSFFPE classifier and the CMSClassifier to the FF-sample data of the CIT cohort (GSE39582) (11).

(v) Application in an independent FFPE-RNA dataset: lastly, we applied the CMSFFPE classifier to the FFPE-RNA application cohort. Molecular features per CMS were assessed by analyzing differential expression of genes (DEG) per CMS compared with the others, and visualized with a heatmap, using the R packages "edgeR" and "pheatmap." Gene set enrichment analyses for samples with read counts >$10^7$ were performed and visualized using the R packages "fgsea" and "pheatmap."

### Analyses of TSO500 mutation data

Fastq data were summarized in standardized combined variant output reports, using the local app v2.2.0 docker container (Illumina) executed on a local HP Unix server via Singularity. Combined variant output reports were parsed to select variants with an amino-acid change, a frameshift, splice variants and start- and stop-events. To account for differences in tumor sample purity, tumor cell percentages, as assessed by a certified pathologist, were used to correct variant allele frequencies. To select for driver variants, the following filter criteria were used: corrected variant allele frequency >0.10 for oncogenes and >0.40 for tumor suppressor genes; presence of a variant in the population ≤$10^{-5}$. To avoid inclusion of assay-specific sequence artifacts, an in-house reference database was set up that included more than 450 TSO500 profiled tumor samples. This reference was used to filter out those variants that occurred in more than 50 percent of the tumors. To avoid missing important somatic mutations, variants that were seen ≥1 cases of the TCGA (from cBioportal) (COAD and READ), were maintained. Also, somatic variants reported in ≥10 cases of the Memorial Sloan Kettering Cancer Center (from cBioportal) (pan cancer) were maintained.

Cohort and sample overviews and visualizations were implemented in the R2 Genomics Analysis and Visualization Platform (https://r2.amc.nl), together with sample annotations. Furthermore, oncoplots were generated to visualize mutations per CMS and per anti-EGFR response group, using the R package "ComplexHeatmaps" from BioConductor (59).

### Associations of subtypes with clinical data

Analyses of clinical data were performed separately from analyses of molecular data, by different researchers. SPSS statistical software, version 28.0.1.1, and R version 4.0.5, were used to perform analyses into clinical correlates of molecular characteristics per tumor. Non-parametric tests were carried out to analyze correlations between continuous clinical data and mutations or CMSs. The Chi-square test, the Fisher's exact test or the Fisher-Freeman-Halton Exact test was used to associate categorical data with molecular characteristics of tumors. Univariable and multivariable logistic regression analyses were performed to identify most significant determinants of anti-EGFR response in the treatment group. Response was defined as the ORR: partial or complete response as registered by the treating physician.

Survival analyses were carried out using the survival package in R. OS was defined as time from diagnosis until death of any cause. Survival curves for molecular and clinical subgroups were generated with the Kaplan-Meier method and compared with the log-rank test. Cox proportional hazards modeling for OS was performed, after confirming the assumption of proportional hazards. Univariable and multivariable analyses were carried out to identify prognostic determinants for OS.

Most significant and most clinically relevant variables from univariable analyses were included in multivariable analyses. Variables with missing values were excluded from multivariable analyses. *P*-values were two-sided with an alpha < 0.05. Multiple testing correction was performed for analyses with sequencing data.

## Data Availability

The CMSFFPE classifier code from this publication has been deposited to github repository https://github.com/yswutan/CMSFFPE. The RNA-sequencing data from this publication have been deposited to the GEO database https://www.ncbi.nlm.nih.gov/geo/query/acc.cgi and assigned the identifier GSE267010.

## Supplementary Information

## Acknowledgements

The authors of this article would like to thank the Netherlands Cancer Registry and PALGA for the dedicated data collection. Furthermore, we would like to express our gratitude toward Illumina for kindly providing TSO500 assays for analysis in this study, and Pilar Ramos and Chris Zwanenburg for advice on data analyses. This work was supported by The New York Stem Cell Foundation (NYSCF-I-R43), grants from the European Research Council (ERC-

CoG 101045612-NIMICRY), ZonMw (Vici 09-15018-21-10029), Dutch Cancer Society grant (KWF 14182), The Oncode Institute Technology Development Fund (P2021-0011), Research Grants Council (Project No. C4024-22GF, R4007-23, 11103921) of the Hong Kong Special Administrative Region, China, and partially sponsored by Shenzhen Bay Scholars Program awarded to X Wang. The funder had no role in study design or manuscript.

## Author Contributions

TR de Back: conceptualization, data curation, software, formal analysis, validation, investigation, visualization, methodology, and writing—original draft.

T Wu: conceptualization, data curation, software, formal analysis, validation, investigation, visualization, methodology, and writing—original draft.

PJM Schafrat: conceptualization and writing—review and editing.

S ten Hoorn: conceptualization, methodology, and writing—review and editing.

M Tan: formal analysis and investigation.

L He: formal analysis and investigation.

SR van Hooff: data curation, formal analysis, investigation, and visualization.

J Koster: data curation, formal analysis, investigation, and visualization.

LE Nijman: investigation.

GR Vink: conceptualization and writing—review and editing.

IJ Beumer: conceptualization and writing—review and editing.

CC Elbers: conceptualization, supervision, funding acquisition, project administration, and writing—review and editing.

KJ Lenos: conceptualization, supervision, funding acquisition, and writing—review and editing.

DW Sommeijer: conceptualization, resources, supervision, funding acquisition, validation, methodology, project administration, and writing—review and editing.

X Wang: conceptualization, resources, supervision, funding acquisition, validation, methodology, project administration, and writing—review and editing.

L Vermeulen: conceptualization, resources, supervision, funding acquisition, validation, methodology, project administration, and writing—review and editing.

## Conflict of Interest Statement

L Vermeulen received consultancy fees from Bayer, MSD, Genentech, Servier, Roche, Novartis, and Pierre Fabre, but these had no relation to the content of this publication. L Vermeulen is currently an employee of Genentech Inc. and shareholder of Roche. GR Vink received consultancy fees from BMS, Merck, Servier, Personal Genome Diagnostics, Bayer, Sirtex, Pierre Fabre, Lilly, and Delfi Diagnostics, but these had no relation to the content of this publication. The other authors declare no potential conflicts of interest.

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
