## [Reviewer comments · Life Science Alliance]

Life Science Alliance

A consensus molecular subtypes classification strategy for clinical colorectal cancer tissues

Tim de Back, Tan Wu, Pascale Schaftrat, Sanne ten Hoorn, Miaomiao Tan, Lingli He, Sander van Hooff, Jan Koster, Lisanne Nijman, Geraldine Vink, Ines Beumer, Clara Elbers, Kristiaan Lenos, Dirkje Sommeijer, Xin Wang, and Louis Vermeulen

DOI: <https://doi.org/10.26508/lsa.202402730>

Corresponding author(s): *Louis Vermeulen, Amsterdam University Medical Centers*

Review Timeline:	Submission Date:	2024-03-20
	Editorial Decision:	2024-03-21
	Revision Received:	2024-05-01
	Editorial Decision:	2024-05-03
	Revision Received:	2024-05-08
	Accepted:	2024-05-09

Transaction Report:

Please note that the manuscript was previously reviewed at another journal and the reports were taken into account in the decision-making process at *Life Science Alliance*.

Referee #1 Review

Remarks for Author:

In this manuscript, de Back et al. developed a method to assess colorectal cancer (CRC) consensus molecular subtypes (CMS) through transcriptomic profiling of FFPE samples. The CMS classification was established in 2015 through the analysis of fresh frozen material. This paper presents a methodological advance compared to the original study, as FFPE material is routinely used in pathology departments. The study is well-performed and the data are convincing. However, it's worth noting (as the authors do) that there have been many efforts to implement the CMS classification in the clinical setting. In the absence of comparative analyses with these other approaches, I am skeptical that the methodology developed in the current study represents a significant advance. Classification methods based on measuring a reduced subset of marker genes in FFPE material (Morris et al., 2021) likely represent an easier approach to bringing the CMS classification to the clinical setting. Determining CMS from either IHC - a method previously developed by the authors (Trinh et al., 2017; Ten Hoon et al., 2018) - or AI-assisted analyses of histological images have shown good specificity and sensitivity (Sirinukunwattana et al., 2021), and appear to be an easier (and cheaper) alternatives to the one developed herein. Beyond this criticism, the current study largely confirms previous observations regarding the features of the different CMS. The data on anti-EGFR therapy, particularly the association between response and driver mutations, are interesting and deserve further validation in additional patient cohorts.

Referee #2 Review

Remarks for Author:

In this manuscript by Tim R. de Back et al., the authors present a strong case for a new CMS-FFPE classifier, to overcome the issues with application of the original RF classifier when used in FFPE tumour tissue. The authors use comparisons for fresh frozen RNA data as their benchmark, indicating that in some cases their new classifier improves classification accuracy. The authors have assembled a comprehensive series of data and characterised these to a very high standard, which will provide the field with additional cohorts for further discovery and validation work.

This study adds to the existing literature of surrogate classifiers that attempt to deliver CMS classification using diagnostic tissue. In doing so, the authors conclude that this CMS-FFPE approach may be essential for clinical implementation of CMS.

I do have some relatively minor questions around the difference between FF and FFPE, in the context of how we should interpret findings from previous studies and indeed how we can learn from the samples that remain consistent v conflicting in terms of their CMS call using either classifier.

Point 1:

For the cases in Discovery that had matched FF and FFPE material, can the authors comment on how spatially "close together" these samples would have been within the tumour mass, particularly when interpreting the data in lines 303-306?

The authors have already described the widespread heterogeneity that exists within a patient sample, so if these FF and FFPE were not precisely matched in terms of location, the use of FF CMS calls as the benchmark would be misleading, given that CMS calls will truly vary at different regions naturally, particularly in the most stromal/inflammatory tumours. Should the true benchmark be the new CMSClassifier calls in FF tissue that has been extracted, if the FF extraction is more closely aligned to the FFPE sample?

This is also relevant for the cohort on line 123 "As an FFPE-RNA validation cohort, a total of 104 FFPE tumor biopsy or resection samples" - the use of biopsy v resection sample introduces another variable that contributes to CMS classification heterogeneity.

Point 2:

Data in line 278-279 that provides evidence for bias towards degradation of immune-specific genes in FFPE v FF is very insightful and offers many explanations for variability of immune-related signatures across cohorts. Can the authors further deconvolute (or at least see if any correlations) in particular types of immune cells/signals that may be most depleted here. Is this confined to some specific transcripts within the immune signature, or indeed lineage-specific immune cells (using the like of MCP or other).

Overall, the approach and data in figure 1 is strong.

Point 3:

In the CIT validation cohort, the authors switch to using "network-based CMS classification labels"; I'm not sure if this is an additional classifier as opposed to staying with the RF classifier?

Point 4:

The biggest change in CMS calls from Figure 2C is the reduction in CMS1/3/4 and increase in CMS2 - can the authors comment on this, and if this is due to selection of more epithelial-specific genes in Figure 1D/E within the classifier? What is it about a CMSClassifier CMS1 or CMS4 that becomes a CMS-FFPE CMS2 that is different to a consistent CMS4?

Perhaps a small analysis using the approaches in Figure 3B to look at the consistent v conflicting CMS called samples; would assume that the same CMS call in both methods would still align to the biological hallmarks, but what is the nature of those that swap?

Point 5:

Line 324 - biopsy and resection samples; were there any specific trends in terms of QC check, degradation and/or sequencing reads overall that were associated with the use of the two different types of clinical sample?

Point 6, Introduction/Discussion:

The authors introduce (lines 82-84) that there are other approaches developed for clinical implementation of CMS; it is incorrect to say that the H&E-based CMS does not deliver specific information on intra-tumoural heterogeneity. This should be clarified.

In addition, the authors should describe how their new system outcompetes these other orthogonal approaches for CMS classification in the Discussion section, particularly those that work well in FFPE material yet avoid the use of transcriptional profiling entirely, primarily IHC-CMS & imCMS. How do we move the field towards more clinical adoption of classifiers like CMS?

Referee #3 Review

Comments on Novelty/Model System for Author:

mixture of publically available data and a new sequencing set. No models used

Remarks for Author:

CMS classification has changed the way the research world thinks about and classifies colorectal cancer, however its clinical take up has been limited, constrained by the limited scope of CRC treatment options. Thus although I think generating an FFPE curated CMS stratification tool is very useful from a research perspective, I think the clinical implications are comparatively limited and thus the focus of the paper on the clinical application is prob distracting.

The team here is excellent and was instrumental in defining the original CMS algorithms, however I have a number of technical and clinical applicability concerns

Technical

1. The manner in which the various patient and sample sets are described and then discussed throughout the manuscript is unclear. Table S1 should be presented as a table in the main body of the manuscript, and shown where Table S1 is currently referenced for transparency.
2. In the methods, the authors describe a different bioinformatic approach to TPM quantification for FF and FFPE derived data and do not provide justification as to why this additional factor potentially contributing to a technical difference was introduced.
3. Table S1 notes that the gene expression data for the Fresh Frozen samples in the discovery cohort were acquired using Affymetrix microarray methods. There has been no mention of this in the main body of the manuscript, and the text does not specify that the "gold standard" CMS classification was determined from a data type different from the gene expression method the authors intend to apply their new classifier to.

4. In the Background section, the authors write:

"Furthermore, the CMSs were predominantly established on high-quality RNA data derived from fresh-frozen (FF) tumor samples, whereas formalin-fixed paraffin-embedded (FFPE) tissue preservation, yielding highly degraded RNA, is standardly used in the clinic. CMS classification on RNA from FFPE tissue using the original CMS classifier is therefore less accurate (22, 23)"

The position where these citations have been placed is misleading, as it implies an evidence base for lower accuracy of the original CMS classifier in FFPE samples compared to matched FF samples, which is not what the cited papers discuss.

In fact, both reference 22 (Gallego Romero et al., 2014) and reference 23 (Sigrgeirsson et al., 2014) do not refer to RNA quality from FFPE material at all, but only talk about degraded RNA based on RIN values from fresh material.

In the absence of existing published evidence (in a like-for-like comparison), the authors have not fully supported their assertion that "CMS classification on RNA from FFPE tissue using the original CMS classifier is therefore less accurate" by failing to demonstrate worse performance of the existing classifiers with FFPE RNAseq data compared to FF RNAseq, or even between FFPE microarray data compared to FF microarray in matched samples. If a matched FF and FFPE sample set was not available with RNA sequencing data for the fresh-frozen samples, then it is unclear why the authors have chosen to proceed with an entirely different gene expression technology for the FFPE samples given that the existing literature shows good concordance of quantification between FF and FFPE sample for gene expression microarrays (as shown, for example, in Wimmer et al., 2018 and many others).

While the authors have made the comparison of classification performance between FFPE and FF samples in their "Validation of the CMSFFPE Classification Model" section of the results, these were all "data not shown", when it is fundamental information to be shown when asserting improved performance of a new method compared to existing methods. Again, given the lack of clarity as to where Microarray data is being used, this should be made clear in both the figures and the text.

5. Given that Table S1 indicates that the gene expression data for the FF samples were acquired by microarray and for the FFPE samples they were acquired by RNA sequencing, then Figure 1A and Figure 1D do not represent appropriate and like-for-like comparisons, as "normalized data" from microarray results and those from RNA-sequencing are not directly comparable due to a multitude of methodological differences, from the manner by which the data are acquired and quantified to the downstream processing of the raw data from each method.

6. The work on the "FFPE-RNA validation cohort" (Figures 3 and 4 and the related Tables and Supplementary Material) does not constitute a validation of the method. This work could potentially be an interesting tangential investigation narrative if the concordance of the CMSFFPE classifier with "gold standard" CMS classification results from matching fresh frozen RNAseq data were available and good performance of their new CMSFFPE classifier was conclusively established. As that material/information is not available for this cohort, all assertions of equivalence based on clinical associations are largely circumstantial, providing insufficient evidence to qualify as validation of the method.

Clinical application

1. I don't really see the point of the addition of the CMS classifier to APC and mutation status in stratifying RAS/RAF wt patients. APC and P53 seq is not undertaken routinely and thus this stratification is not realistically achievable clinically
2. In terms of clinical applicability the most useful comparison would be the FFPE classifier against image based CMS (Sirinukunwattana, Gut 2021) this is the comparison that probably matters most - This could be easily achieved by comparing the performance of each tool in the TCGA data which has been analysed in both papers
3. Another useful thing to do would be to see how representative FFPE CMS on single biopsy is against whole tumour CMS. How accurate is CMS on biopsy at predicting whole tumour CMS?. A useful dataset for this would be the BOSS study (Ubink et al., 2017)
4. The peritoneal mets conclusion is an overreach - not even close to a significant difference between CMS2 and CMS4

Minor

Throughout. - Golden standard should be gold standard

March 21, 2024

Re: Life Science Alliance manuscript #LSA-2024-02730-T

Prof. Louis Vermeulen
Amsterdam UMC
Meibergdreef 9
Amsterdam, Noord-Holland 1105 AZ
NETHERLANDS

Dear Dr. Vermeulen,

Thank you for submitting your manuscript entitled "A consensus molecular subtypes classification strategy for clinical colorectal cancer tissues" to Life Science Alliance. We invite you to submit a revised manuscript addressing the following Reviewer comments:

- Address Reviewer 2's comments.
- Address Reviewer 3's comments, except for the Clinical application points #1-3.

Thank you for this interesting contribution to Life Science Alliance. We are looking forward to receiving your revised manuscript.

Sincerely,

- A letter addressing the reviewers' comments point by point.
- An editable version of the final text (.DOC or .DOCX) is needed for copyediting (no PDFs).
- High-resolution figure, supplementary figure and video files uploaded as individual files: See our detailed guidelines for preparing your production-ready images, <https://www.life-science-alliance.org/authors>
- Summary blurb (enter in submission system): A short text summarizing in a single sentence the study (max. 200 characters including spaces). This text is used in conjunction with the titles of papers, hence should be informative and complementary to the title and running title. It should describe the context and significance of the findings for a general readership; it should be written in the present tense and refer to the work in the third person. Author names should not be mentioned.
- By submitting a revision, you attest that you are aware of our payment policies found here: <https://www.life-science-alliance.org/copyright-license-fee>

B. MANUSCRIPT ORGANIZATION AND FORMATTING:

Rebuttal document - De Back et al. - Transferred manuscript LSA-2024-02730-T to Life Science Alliance

We thank the editors and reviewers for their thoughtful comments on our manuscript. We very much appreciated the suggestions and recommendations that helped us to significantly improve the study. With the adaptations and substantial amount of novel data and analyses we provide, we have addressed all points and hope that you will consider the manuscript suitable for publication in *Life Science Alliance*.

Please find below our point-by-point reply. Please also note that we have included the unedited text of the reviewers, but in places we have split and renumbered the queries for clarity.

Referee 2

In this manuscript by Tim R. de Back et al., the authors present a strong case for a new CMS-FFPE classifier, to overcome the issues with application of the original RF classifier when used in FFPE tumour tissue. The authors use comparisons for fresh frozen RNA data as their benchmark, indicating that in some cases their new classifier improves classification accuracy. The authors have assembled a comprehensive series of data and characterised these to a very high standard, which will provide the field with additional cohorts for further discovery and validation work.

This study adds to the existing literature of surrogate classifiers that attempt to deliver CMS classification using diagnostic tissue. In doing so, the authors conclude that this CMS-FFPE approach may be essential for clinical implementation of CMS.

I do have some relatively minor questions around the difference between FF and FFPE, in the context of how we should interpret findings from previous studies and indeed how we can learn from the samples that remain consistent v conflicting in terms of their CMS call using either classifier.

We would like to thank the referee for appreciating the merit of our study. We are happy to answer the questions raised by the referee, as we agree these are relevant and interesting points that need some further elucidation.

Point 1:

For the cases in Discovery that had matched FF and FFPE material, can the authors comment on how spatially "close together" these samples would have been within the tumour mass, particularly when interpreting the data in lines 303-306?

The authors have already described the widespread heterogeneity that exists within a patient sample, so if these FF and FFPE were not precisely matched in terms of location, the use of FF CMS calls as the benchmark would be misleading, given that CMS calls will truly vary at different regions naturally, particularly in the most stromal/inflammatory tumours. Should the true benchmark be the new CMSClassifier calls in FF tissue that has been extracted, if the FF extraction is more closely aligned to the FFPE sample?

This is also relevant for the cohort on line 123 "As an FFPE-RNA validation cohort, a total of 104 FFPE tumor biopsy or resection samples" - the use of biopsy v resection sample introduces another variable that contributes to CMS classification heterogeneity.

We agree with the referee that spatial differences between the matched FF and FFPE resection specimens of the Discovery cohort could substantially impact CMS calls due to intra-tumor (CMS) heterogeneity, as for example shown in (1) and (2). Unfortunately, we were not able to reconstruct the spatial relation between the FF and FFPE samples. However, we did compare the CMS calls of the CMSFFPE classifier on FF data with the calls on FFPE data, and found a significant

concordance between FF and FFPE calls of the same tumor, as described in lines 171-174, and shown in Figure R1-D. Although this does not rule out bias in CMS calls due to spatial distance between FF and FFPE specimens, it does indicate consistency in CMS calls from the CMSFFPE classifier, even if samples were spatially apart. We added Figure R1 to the Supplements to show these data (included in manuscript as Figure S5).

At the same time, we would like to point out that a strong correlation between FF benchmark CMS calls and FFPE CMSFFPE calls was found, despite the widespread intra-tumor heterogeneity of CRC, as shown in Figure R1-A. Spatial distance between the FF and FFPE specimens would probably underestimate the accuracy of the new CMSFFPE classifier, which would only improve when FF and FFPE samples are in close proximity within a tumor, or in fact the same region.

We agree with the referee that intra-tumor heterogeneity could also impact the CMS calls of biopsy specimens in the FFPE-RNA application cohort (previously FFPE-RNA validation cohort), as biopsies cover a relatively small area of the tumor, hence more dependent on sampling region. We have chosen to include both biopsies and resection specimens to most accurately reflect the clinical setting. Despite including a substantial number of biopsies (n=35, Table 2), we were still able to recover all key associations of the CMSs in this cohort. We would therefore argue that CMSFFPE calls on biopsies did not hinder the detection of the true biological CMS subgroups.

We have added a discussion on the impact of sampling region within a tumor on CMS calls in the Limitations section of the Discussion, lines 368-385.

Figure R1. Accuracy and FFPE-FF label concordance per CMS classifier.

(A-C) Classifier accuracies against benchmark core labels derived from (3), for the CMSFFPE classifier (A), the original random forest CMSClassifier (B), and the single-sample predictor (C). (D-F) The concordance of CMS calls on FFPE-derived RNA-sequencing data versus FF-derived microarray data for the CMSFFPE classifier (D), the original random forest CMSClassifier (E), and the single-sample predictor (F). P values were calculated with Binomial Exact tests. FFPE, formalin-fixed paraffin-embedded; MA, microarray; FF, fresh-frozen; RF, random forest; SSP, single-sample predictor.

Point 2:

Data in line 278-279 that provides evidence for bias towards degradation of immune-specific genes in FFPE v FF is very insightful and offers many explanations for variability of immune-related signatures across cohorts. Can the authors further deconvolute (or at least see if any

correlations) in particular types of immune cells/signals that may be most depleted here. Is this confined to some specific transcripts within the immune signature, or indeed lineage-specific immune cells (using the like of MCP or other).

Overall, the approach and data in figure 1 is strong.

We thank the referee for the interest in these data. As suggested by the referee, we have compared the degradation level of different (immune) cell lineages in the low-quality and moderate-quality discovery cohorts, and the application cohort, using the MCP algorithm (Figure R2, see below). We found that the degradation level of immune cells is overall higher compared to fibroblasts (reflected by a lower TIN-score). Interestingly, we indeed observed lineage-specific variations in transcript degradation as the reviewer proposed. For example, RNA from monocytes and neutrophils is generally better preserved in FFPE tissue compared to RNA from NK cells, dendritic cells or cytotoxic T lymphocytes. This finding was consistent across the three profiled cohorts with varying data quality (indicated with a red dotted line representing the median TIN-score of all genes in the dataset). We would therefore argue that the degradation of immune cells is cell-type specific and not a general feature of all immune cells. This suggests that the detection of immune subtypes is not necessarily hindered by FFPE preservation, as some immune cell types are relatively well preserved. We have added this Figure to the Supplements of the manuscript (Figure S4) and mentioned the key conclusion in the Results section, lines 146-147.

Figure R2. Comparison of TIN-score per gene sorted by MCP cell types. (A) TIN-score comparison per gene per cell type included in the MCP algorithm in the low-quality discovery cohort, (B) the moderate-quality discovery cohort, (C) and the FFPE-RNA application cohort. The red dotted lines indicate the median TIN-score of all genes in the respective dataset. TIN, transcript integrity number; FFPE, formalin-fixed paraffin-embedded.

Point 3:

In the CIT validation cohort, the authors switch to using "network-based CMS classification labels"; I'm not sure if this is an additional classifier as opposed to staying with the RF classifier?

We apologize for the confusion. These labels have not been derived from an additional classifier. To accurately compare the performance of the original RF CMSClassifier with the new CMSFFPE classifier, we used the core CMS labels from the CIT cohort as provided in the study by Guinney et al. (3). In the discovery phase of this paper, a network analysis was performed with the subtype labels from six original transcriptomic classification systems to detect which labels clustered most closely together, identifying four subgroups, termed the CMSs (core CMS calls). These represent the benchmark labels for the CMSs in this primary cohort. Next, the original RF CMSClassifier was built to classify remaining samples. For our analyses of the CIT cohort, we exclusively used core CMS calls from the discovery phase of the paper as a benchmark (termed network-based CMS classification labels). By doing so, we could directly compare the accuracy of the original RF CMSClassifier with the CMSFFPE classifier on these data. To prevent further confusion, we explained this in more detail in the Methods section, lines 431-435, and Results section, lines 166-168, 183-185, and we changed the term 'network-based CMS classification labels' to core CMS calls.

Point 4:

The biggest change in CMS calls from Figure 2C is the reduction in CMS1/3/4 and increase in CMS2 - can the authors comment on this, and if this is due to selection of more epithelial-specific genes in Figure 1D/E within the classifier? What is it about a CMSClassifier CMS1 or CMS4 that becomes a CMS-FFPE CMS2 that is different to a consistent CMS4?

Perhaps a small analysis using the approaches in Figure 3B to look at the consistent v conflicting CMS called samples; would assume that the same CMS call in both methods would still align to the biological hallmarks, but what is the nature of those that swap?

We thank the referee for pointing this out. Indeed, part of the CMS1/3/4 tumors, as determined by the RF CMSClassifier, seems to be classified as CMS2 by the CMSFFPE classifier, based on Figure 2C. From Figure 2B, it becomes clear that indeed the accuracy of the new CMSFFPE classifier to detect CMS2 is higher than from the original RF CMSClassifier, compared to core CMS calls of the CIT cohort (93.1% versus 81.5%). On the other hand, the accuracy to detect CMS1/3/4 with the CMSFFPE classifier is slightly lower compared to the original RF CMSClassifier. This correlates with the shift from CMS1/3/4 tumors predicted by the RF classifier to CMS2 by the CMSFFPE classifier. To further explain this, we checked the CMS probability scores of the CMSFFPE classifier per CMS in the CIT cohort and plotted them against the original CMSClassifier labels (Figure R3, see below). We found that the probability scores of the samples that switched from CMS call (discordant samples) were significantly lower compared to the samples that had concordant CMS calls between the CMSClassifier and the CMSFFPE classifier (Figure R3). This indicates that these samples have a more CMS-mixed phenotype, which affected the classification. We added this figure to the supplements (Figure S6) and briefly discussed these findings in the Results section, lines 189-193.

Figure R3. CMSFFPE prediction scores compared to CMSClassifier calls CIT cohort (A-D) Comparison of the concordantly and discordantly classified samples (CMSFFPE classifier versus original CMSClassifier), based on CMSFFPE prediction score for CMS1 (A), CMS2 (B), CMS3 (C) and CMS4 (D). P values indicate comparison of prediction score between concordantly and discordantly classified samples (Wilcoxon signed rank test).

Point 5:

Line 324 - biopsy and resection samples; were there any specific trends in terms of QC check, degradation and/or sequencing reads overall that were associated with the use of the two different types of clinical sample?

This is indeed an interesting point, that deserves further attention. We have checked the TIN-scores, insert sizes and uniquely mapped reads of biopsies and compared it to resections

specimens in the FFPE-application cohort, and found a higher RNA quality of biopsy specimens versus resection specimens (Figure R4). The difference in RNA quality between biopsies and resections is most likely due to a longer sample processing time of resection specimens, as time to fixation (cold ischemia time) is instrumental to RNA quality (4); the biopsy procedure is generally quicker and biopsies require less sample preprocessing prior to fixation. The RNA quality of both biopsies and resection specimens in the FFPE-application cohort was sufficient for CMS classification. We mentioned this comparison in the Results section, lines 216-218, and included the Figure R4 in the supplements as Figure S8.

Figure R4. Comparison of data quality between biopsies and resection specimens. (A) TIN-score, (B) median insert size and (C) uniquely mapped reads from biopsies and resection specimens. P values comparing biopsies with resection specimens originate from the Wilcoxon signed rank test.

Point 6. Introduction/Discussion:

The authors introduce (lines 82-84) that there are other approaches developed for clinical implementation of CMS; it is incorrect to say that the H&E-based CMS does not deliver specific information on intra-tumoural heterogeneity. This should be clarified.

In addition, the authors should describe how their new system outcompetes these other orthogonal approaches for CMS classification in the Discussion section, particularly those that work well in FFPE material yet avoid the use of transcriptional profiling entirely, primarily IHC-CMS & imCMS. How do we move the field towards more clinical adoption of classifiers like CMS?

We thank the referee for pointing this out. We corrected the statement in lines 96-98 that H&E-based CMS detection does not deliver information on intra-tumoural heterogeneity, as it is indeed able to detect regions with predominantly epithelial tumor cells, stromal infiltration or immune invasion, and multi-region CMS classification would be feasible.

We agree that previously developed alternative classification strategies, such as IHC-based and image-based CMS classification, are quick, easily clinically adopted and cost-effective. However, we would argue that the major advantage of the CMSFFPE classifier lies in the acquisition of transcriptome-wide data from readily available clinical tumor samples, instead of acquiring only a subset of data. We feel that this is of significant importance for the CRC research field, enabling reuse of transcriptomic data for future projects. This could also be of importance for future clinical adoption of the CMSFFPE classifier, as it would allow for multi-classification of one tumor with

different CRC transcriptome taxonomies, assessment of additional therapeutic targets beyond the predictive value of the CMSs, and incorporation of novel developments. We do agree that the initial utility of the CMSFFPE classifier probably lies mostly within research, and we have changed the manuscript accordingly.

Next-generation sequencing has become quicker and more affordable (5, 6), and ready-to-use bioinformatics platforms have been developed to ease discovery and classification of complex transcriptomic data (7-9). To move the field forward to clinical adoption of classifiers like the CMSs, we would argue that standardized FFPE sequencing and classification pipelines should be developed, as we aimed to do within this project. Furthermore, clinical decision-making based on transcriptomic data should be further refined, and could probably be discerned most valuably by means of transcriptome-wide data. We discussed this more elaborately in the Discussion section, lines 348-360.

Referee 3

CMS classification has changed the way the research world thinks about and classifies colorectal cancer, however its clinical take up has been limited, constrained by the limited scope of CRC treatment options. Thus although I think generating an FFPE curated CMS stratification tool is very useful from a research perspective, I think the clinical implications are comparatively limited and thus the focus of the paper on the clinical application is prob distracting.

The team here is excellent and was instrumental in defining the original CMS algorithms, however I have a number of technical and clinical applicability concerns.

We would like to thank the referee for the valuable comments and suggestions to our manuscript. We agree that the initial value of the CMSFFPE classifier probably lies mostly within research, and therefore, we have changed the focus of the paper accordingly.

Technical

1. The manner in which the various patient and sample sets are described and then discussed throughout the manuscript is unclear. Table S1 should be presented as a table in the main body of the manuscript, and shown where Table S1 is currently referenced for transparency.

We apologize for the unclarity regarding the patient and samples sets. We have updated the methods and results sections to more clearly indicate the specific set used. Furthermore, we have added Table S1 to the main manuscript as Table 1, as suggested by the referee. This indeed improves transparency.

2. In the methods, the authors describe a different bioinformatic approach to TPM quantification for FF and FFPE derived data and do not provide justification as to why this additional factor potentially contributing to a technical difference was introduced.

We thank the referee for drawing our attention to this point. We indeed used both RSEM and featureCounts for TPM quantification. For the FFPE samples of the discovery cohort, we utilized STAR for mapping and RSEM for TPM calculation, similar to the training set from TCGA. By doing so, the establishment of the CMSFFPE model does not suffer from any additional bias due to differences in TPM calculation methodology between the discovery cohort and the training cohort (TCGA). As for the FFPE application cohort, we also employed STAR for mapping and featureCounts for quantification, since this was recommended by the sequencing company (<https://www.takarabio.com/products/next-generation-sequencing/bioinformatics-tools/cogent-ngs-analysis-pipeline>). Essentially, TPM quantification using these two bioinformatic approaches follows the same formula:

$$\text{TPM} = \frac{(\text{Reads mapped to gene}/\text{gene length}) \times 10^6}{\sum(\text{Reads mapped to gene}/\text{gene length})}$$

We used TPM to quantify gene expression levels, since it normalizes gene expression levels for differences in sequencing depth between samples. By scaling expression levels to a common denominator of one million and dividing expression counts by transcript length, TPM allows for direct comparison of gene expression levels across samples with varying sequencing depths. Furthermore, a previous study has compared the TPM estimated by RSEM and featureCounts, and showed that these methods are highly concordant (Figure 2A in (10)). Hence, we would argue

that the two approaches for TPM quantification introduce limited bias. We have also addressed this more thoroughly in the Methods section, lines 495-498, 500-502, 506-507.

3. *Table S1 notes that the gene expression data for the Fresh Frozen samples in the discovery cohort were acquired using Affymetrix microarray methods. There has been no mention of this in the main body of the manuscript, and the text does not specify that the "gold standard" CMS classification was determined from a data type different from the gene expression method the authors intend to apply their new classifier to.*

We fully agree with the referee that this should be more clearly noted in the main body of the manuscript. We added this to the methods, lines 422-423, 430, and results, lines 139, 142-143, 167, 182 and included this in the Limitations section of our study, lines 363-368. We have also stated this more clearly in the Figures and Figure legends. We specifically thank the reviewer for pointing out this critical issue. Sometimes, when dealing with these datasets for years, these key features are taken for granted.

4. *In the Background section, the authors write:*

"Furthermore, the CMSs were predominantly established on high-quality RNA data derived from fresh-frozen (FF) tumor samples, whereas formalin-fixed paraffin-embedded (FFPE) tissue preservation, yielding highly degraded RNA, is standardly used in the clinic. CMS classification on RNA from FFPE tissue using the original CMS classifier is therefore less accurate (22, 23)"

The position where these citations have been placed is misleading, as it implies an evidence base for lower accuracy of the original CMS classifier in FFPE samples compared to matched FF samples, which is not what the cited papers discuss.

In fact, both reference 22 (Gallego Romero et al., 2014) and reference 23 (Sigurgeirsson et al., 2014) do not refer to RNA quality from FFPE material at all, but only talk about degraded RNA based on RIN values from fresh material.

We apologize for the confusion that resulted from our reference placing. We agree that with the current reference positioning it seems as though the studies demonstrate less accurate CMS calling on degraded RNA derived from FFPE tissues, which is indeed not what the references demonstrate. The studies demonstrate that low-quality RNA, as measured with RIN values, obtained from fresh material, significantly impacts downstream analyses of RNA-seq data, such as gene coverage, false positives in differential gene expression and quantification of duplicate reads. As FFPE-derived RNA typically has very low RIN values (generally below 2.5), downstream gene expression analyses are most likely affected in a likewise manner, although this is hypothesis-generating. Some direct evidence is available on the accuracy of cancer subtyping on degraded FFPE-derived RNA compared to FF-derived RNA. One small study performed a comparison between four pairs of FF and FFPE glioblastoma samples that went for RNA-sequencing, and showed that samples with the highest level of degradation cannot be accurately classified (1/4, 25%) (11). Another study shows that more recently FFPE-preserved samples (<4 years) and a higher read depth positively affect subtyping concordance between matched FF and FFPE breast cancer samples, with an overall concordance rate of 54.5% for matched samples sequenced with higher read depth, versus 50% for matched samples sequenced with lower read depth (12). Although these studies are small and not focused on CMS classification of CRC samples, they do provide ground to state that transcriptomic classification on FFPE-derived RNA-data is less accurate than FF-derived RNA data. We have changed the cited passage and replaced references to correct and improve the evidence base for this notion (lines 98-105).

In the absence of existing published evidence (in a like-for-like comparison), the authors have not fully supported their assertion that "CMS classification on RNA from FFPE tissue using the original CMS classifier is therefore less accurate" by failing to demonstrate worse performance of the existing classifiers with FFPE RNAseq data compared to FF RNAseq, or even between FFPE microarray data compared to FF microarray in matched samples. If a matched FF and FFPE sample set was not available with RNA sequencing data for the fresh-frozen samples, then it is unclear why the authors have chosen to proceed with an entirely different gene expression technology for the FFPE samples given that the existing literature shows good concordance of quantification between FF and FFPE sample for gene expression microarrays (as shown, for example, in Wimmer et al., 2018 and many others).

As discussed under point 4 of referee 3, there is some literature evidence available that the accuracy of cancer subtyping using RNA-seq data is affected by FFPE-preservation, as compared to FF-preservation, although this has indeed not been shown for CMS classification on CRC tissue. However, we do show in our data in Figure 2A and Figure R1 (referee 2 – point 1) that the original RF CMSClassifier performs poorly in two degraded FFPE-datasets (RNA-seq data, accuracies 40.9% and 66.7%), with FF benchmark labels, although originating from microarray analysis. We have also shown that FFPE-preservation impacts RNA degradation in a cell-specific manner (Figure R2), for example high preservation of fibroblast genes versus low preservation of NK cells, which is expected to impact classification accuracy if not taken into account, which is confirmed by the low accuracy of the original CMSClassifier in the low-quality Discovery set (Figure R1).

Interestingly, the previously mentioned study in breast cancer patients also evaluated subtype concordance between microarray and RNA-seq platforms using patient-matched samples and found a very high subtype consistency (91%), suggesting that the introduced bias by comparing microarray data with RNA-seq data in terms of subtyping calls is limited (12). The study mentioned by the referee indeed shows high gene expression concordance between FF and FFPE microarrays, and at the same time shows high concordance between FFPE array and fresh RNA seq data (Figure 6F) (13). These studies suggest that using FF microarray benchmark labels for FFPE RNA-seq data introduces limited confounding. Nevertheless, we do acknowledge that differences in platforms used between the datasets does add heterogeneity in the comparisons made.

Our main aim was to optimize the CMSClassifier for degraded FFPE-RNA derived from RNA-seq, as the mainstay of current transcriptomic profiling efforts. Unfortunately, for our discovery set the matched FF samples were not available for additional RNA-sequencing. Therefore, we decided to proceed by comparing FF microarray data with FFPE RNAseq data. As abovementioned studies indicate high concordance of gene expression analysis and cancer subtyping between microarray and RNA-seq, we would argue that the introduced bias is limited. We have added the use of different gene expression platforms in the discovery cohort to the Limitations of the Discussion section, lines 363-368.

While the authors have made the comparison of classification performance between FFPE and FF samples in their "Validation of the CMSFFPE Classification Model" section of the results, these were all "data not shown", when it is fundamental information to be shown when asserting improved performance of a new method compared to existing methods. Again, given the lack of clarity as to where Microarray data is being used, this should be made clear in both the figures and the text.

We thank the referee for pointing this out. We have now included all data mentioned in this section in Figure S5 of the Supplements to improve clarity and transparency (Figure R1, see referee 2 –

point 1). We have also indicated in the text and Figures where microarray data was being used, as discussed under point 3 of referee 3.

5. Given that Table S1 indicates that the gene expression data for the FF samples were acquired by microarray and for the FFPE samples they were acquired by RNA sequencing, then Figure 1A and Figure 1D do not represent appropriate and like-for-like comparisons, as "normalized data" from microarray results and those from RNA-sequencing are not directly comparable due to a multitude of methodological differences, from the manner by which the data are acquired and quantified to the downstream processing of the raw data from each method.

We agree with the referee that Figure 1A and Figure 1D do not represent like-for-like comparisons. Even though this is true, studies exist that show high concordance between microarray and RNA-seq gene expression analyses from the same tumor (12, 13), suggesting that the bias introduced by comparing these different methodologies is limited. This is also supported by the fact that the correlation between FF versus FFPE gene expression data is rather high (Figures 1A and 1D), despite the fact that different methodologies were used (microarray versus RNA-seq). Moreover, we found a high concordance between CMS calls of the CMSFFPE classifier on FFPE RNA-seq data versus FF microarray data (Figure R1-D, referee 2 – point 1). However, to clarify that these are not like-for-like comparisons, and rather different methods and tissues, we indicated the different methodologies used in these Figures. We also included this limitation in the Discussion section, lines 363-368.

6. The work on the "FFPE-RNA validation cohort" (Figures 3 and 4 and the related Tables and Supplementary Material) does not constitute a validation of the method. This work could potentially be an interesting tangential investigation narrative if the concordance of the CMSFFPE classifier with "gold standard" CMS classification results from matching fresh frozen RNAseq data were available and good performance of their new CMSFFPE classifier was conclusively established. As that material/information is not available for this cohort, all assertions of equivalence based on clinical associations are largely circumstantial, providing insufficient evidence to qualify as validation of the method.

We agree with the referee that additional validation of the CMSFFPE classifier in another cohort with matched FF tissue stemming from RNA-seq would be desirable. However, we would argue that the key molecular and clinical features of the CMSs, as recovered in this independent FFPE RNA-seq cohort, indicate robust detection of the CMSs using the novel classifier, providing some evidence to state that the classifier properly assigns CMS-labels. Since further validation would be required, we changed the scope of this section from validation of the classifier to application of the novel classifier in an independent cohort (FFPE-RNA application cohort). We toned down the wording that we have validated the classifier and also added a statement to the Discussion section, indicating that additional validation of the CMSFFPE classifier in RNA-seq datasets from patient-matched FF and FFPE samples is needed (lines 404-405).

Clinical application

4. The peritoneal mets conclusion is an overreach - not even close to a significant difference between CMS2 and CMS4

We agree with the referee and removed this conclusion from the manuscript.

Minor

Throughout. - Golden standard should be gold standard

We thank the referee for pointing this out. We have chosen to rephrase this to benchmark labels.

References

1. Dunne PD, McArt DG, Bradley CA, O'Reilly PG, Barrett HL, Cummins R, et al. Challenging the Cancer Molecular Stratification Dogma: Intratumoral Heterogeneity Undermines Consensus Molecular Subtypes and Potential Diagnostic Value in Colorectal Cancer. *Clin Cancer Res.* 2016;22(16):4095-104.
2. Marisa L, Blum Y, Taieb J, Ayadi M, Pilati C, Le Malicot K, et al. Intratumor CMS Heterogeneity Impacts Patient Prognosis in Localized Colon Cancer. *Clin Cancer Res.* 2021;27(17):4768-80.
3. Guinney J, Dienstmann R, Wang X, de Reynies A, Schlicker A, Soneson C, et al. The consensus molecular subtypes of colorectal cancer. *Nature Medicine.* 2015;21(11):1350-6.
4. Compton CC, Robb JA, Anderson MW, Berry AB, Birdsong GG, Bloom KJ, et al. Preanalytics and Precision Pathology: Pathology Practices to Ensure Molecular Integrity of Cancer Patient Biospecimens for Precision Medicine. *Arch Pathol Lab Med.* 2019;143(11):1346-63.
5. Goodwin S, McPherson JD, McCombie WR. Coming of age: ten years of next-generation sequencing technologies. *Nat Rev Genet.* 2016;17(6):333-51.
6. Satam H, Joshi K, Mangrolia U, Waghoo S, Zaidi G, Rawool S, et al. Next-Generation Sequencing Technology: Current Trends and Advancements. *Biology (Basel).* 2023;12(7).
7. Ahmaderaghi B, Amirkhah R, Jackson J, Lannagan TRM, Gilroy K, Malla SB, et al. Molecular Subtyping Resource: a user-friendly tool for rapid biological discovery from transcriptional data. *Dis Model Mech.* 2022;15(3).
8. Hait TA, Maron-Katz A, Sagir D, Amar D, Ulitsky I, Linhart C, et al. The EXPANDER Integrated Platform for Transcriptome Analysis. *J Mol Biol.* 2019;431(13):2398-406.
9. Reyes ALP, Silva TC, Coetzee SG, Plummer JT, Davis BD, Chen S, et al. GENAVi: a shiny web application for gene expression normalization, analysis and visualization. *BMC Genomics.* 2019;20(1):745.
10. Zhao S, Xi L, Zhang B. Union Exon Based Approach for RNA-Seq Gene Quantification: To Be or Not to Be? *PLoS One.* 2015;10(11):e0141910.
11. Esteve-Codina A, Arpi O, Martinez-Garcia M, Pineda E, Mallo M, Gut M, et al. A Comparison of RNA-Seq Results from Paired Formalin-Fixed Paraffin-Embedded and Fresh-Frozen Glioblastoma Tissue Samples. *PLoS One.* 2017;12(1):e0170632.
12. Jovanovic B, Sheng Q, Seitz RS, Lawrence KD, Morris SW, Thomas LR, et al. Comparison of triple-negative breast cancer molecular subtyping using RNA from matched fresh-frozen versus formalin-fixed paraffin-embedded tissue. *BMC Cancer.* 2017;17(1):241.
13. Wimmer I, Troscher AR, Brunner F, Rubino SJ, Bien CG, Weiner HL, et al. Systematic evaluation of RNA quality, microarray data reliability and pathway analysis in fresh, fresh frozen and formalin-fixed paraffin-embedded tissue samples. *Sci Rep.* 2018;8(1):6351.

May 3, 2024

RE: Life Science Alliance Manuscript #LSA-2024-02730-TR

Prof. Louis Vermeulen
Amsterdam University Medical Centers
Meibergdreef 9
Amsterdam, Noord-Holland 1105 AZ
Netherlands

Dear Dr. Vermeulen,

Thank you for submitting your revised manuscript entitled "A consensus molecular subtypes classification strategy for clinical colorectal cancer tissues". We would be happy to publish your paper in Life Science Alliance pending final revisions necessary to meet our formatting guidelines.

- please be sure that the authorship listing and order is correct
- please incorporate any points from the Conclusion section into the Discussion, we only allow a Discussion section
- please update the Data Availability statement with the RNA-seq accession number
- please move the figure legends to after the References list
- please add ORCID ID for corresponding author--you should have received instructions on how to do so
- please add callouts for each panel of the Supplemental Figures into the text

A. FINAL FILES:

B. MANUSCRIPT ORGANIZATION AND FORMATTING:

**Submission of a paper that does not conform to Life Science Alliance guidelines will delay the acceptance of your

manuscript.**

The license to publish form must be signed before your manuscript can be sent to production. A link to the electronic license to publish form will be available to the corresponding author only. Please take a moment to check your funder requirements.

Sincerely,

May 9, 2024

RE: Life Science Alliance Manuscript #LSA-2024-02730-TRR

Prof. Louis Vermeulen
Amsterdam University Medical Centers
Meibergdreef 9
Amsterdam, Noord-Holland 1105 AZ
Netherlands

Dear Dr. Vermeulen,

Thank you for submitting your Research Article entitled "A consensus molecular subtypes classification strategy for clinical colorectal cancer tissues". It is a pleasure to let you know that your manuscript is now accepted for publication in Life Science Alliance. Congratulations on this interesting work.

DISTRIBUTION OF MATERIALS:

Again, congratulations on a very nice paper. I hope you found the review process to be constructive and are pleased with how the manuscript was handled editorially. We look forward to future exciting submissions from your lab.

Sincerely,
